Brief Communication

# Protein destabilization underlies pathogenic missense mutations in ARID1B

Fanny Mermet-Meillon[1,4], Samuele Mercan[1,4], Beatrice Bauer-Probst[1], Cyril Allard[2], Melusine Bleu[1], Keith Calkins[1], Judith Knehr[3], Marc Altorfer [3], Ulrike Naumann [3], Kathleen Sprouffske [1], Louise Barys[1], Fabian Sesterhenn [3,5] ✉ & Giorgio G. Galli [1,5] ✉

ARID1B is a SWI/SNF subunit frequently mutated in human Coffin–Siris syndrome (CSS) and it is necessary for proliferation of *ARID1A* mutant cancers. While most CSS *ARID1B* aberrations introduce frameshifts or stop codons, the functional consequence of missense mutations found in *ARID1B* is unclear. We here perform saturated mutagenesis screens on ARID1B and demonstrate that protein destabilization is the main mechanism associated with pathogenic missense mutations in patients with Coffin–Siris Syndrome.

SWI/SNF is a family of chromatin remodeling complexes. Due to the combinatorial and modular assembly of their subunits, SWI–SNF complexes can assemble in different configurations. The three main variants identified are BAF, pBAF and ncBAF. ARID1A and ARID1B subunits are mutually exclusive subunits of the BAF complex.

*ARID1B* is the most frequently mutated SWI/SNF subunit in the autism-spectrum disorder Coffin–Siris syndrome (CSS)[1,2]. ARID1B-containing BAF complex is critical during neural crest cell differentiation[3]. *ARID1A* instead is the most frequently mutated subunit in cancer[4]. In *ARID1A*-deficient cancers, ARID1B-containing complex sustains BAF complex activity[5,6], explaining the dependency of *ARID1A* mutant cancers on ARID1B (ref. 7). While most CSS mutations in *ARID1B* are loss of function nonsense mutations, the functional relevance of missense mutations is largely unexplored (Fig. 1a, Extended Data Fig. 1a,b and Supplementary Table 1).

Recent cryo-EM studies have explained the structure of ARID1A-containing BAF complex[8,9]. In such structures, only the C-terminal EHD2 domain is visible[8,9]. This domain adopts an all-helical armadillo repeat fold and it is centrally located in the base module of the complex engaging multiple subunits in protein–protein interactions to stabilize the base module structure[8,9]. ARID1A EHD2 domain interacts with a long helical domain of BRG1 (HSA domain), the catalytic subunit of the complex. In vitro, mutations in BRG1 aiming to disrupt the interaction with the ARID1A EHD2 domain have been reported to decrease nucleosome remodeling[9] similar to assembly of an ARID-less BAF complex[8]. However, the in vivo functional consequences of missense ARID1B mutations remain unexplored.

To systematically characterize the ARID1B EHD2 domain structure–function relationship, we performed deep mutational scanning (DMS) with two functional assays: (1) a cellular proliferation assay in which expression of ectopic ARID1B cDNA rescues the lethal phenotype induced by silencing endogenous ARID1B in an *ARID1A* mutant cancer cell line, (2) a sensor assay for protein stability and/or abundance using a bicistronic vector expressing green fluorescent protein (GFP)-tagged ARID1B EHD2 domain and a mCherry protein to normalize expression levels (Fig. 1b) similar to VAMP-seq[10]. We refer to protein stability in a broad sense, including protein misfolding and other events causing rapid clearance through the cellular quality control machinery. We validated these two assays using either wild-type sequences or a construct bearing a deletion in the BC-box, which was previously shown to be critical for protein stability[11] and consequently also for proliferation (Extended Data Fig. 2a,b). We then created plasmid libraries including a total of 8,960 single-residue variants (divided into six pools, Extended Data Fig. 2c). We accurately quantitated the abundance of each allele from next-generation sequencing (NGS) data (Methods and Extended Data Fig. 3a) and, after assessing homogeneous representation of the variants in the different pools (Extended Data Fig. 3b and Supplementary Table 2), we screened two pools (Pool4 and Pool5) encompassing the region of ARID1B at the interface with BRG1 (amino acids 1970–2130) in both cancer proliferation assay and protein stability assay (Fig. 1b and Extended Data Fig. 2c).

We computed differential representation of each variant allele in each screen and observed that most mutations do not affect protein stability or proliferation (Extended Data Fig. 4a,b and Supplementary

[1]Disease Area Oncology, Novartis Biomedical Research, Basel, Switzerland. [2]Disease Area Immunology, Novartis Biomedical Research, Basel, Switzerland. [3]Discovery Sciences, Novartis Biomedical Research, Basel, Switzerland. [4]These authors contributed equally: Fanny Mermet-Meillon, Samuele Mercan. [5]These authors jointly supervised this work: Fabian Sesterhenn, Giorgio G. Galli. ✉e-mail: fabian.sesterhenn@novartis.com; giorgio.galli@novartis.com

Table 3). Direct comparison of the results from the two screening readouts revealed a significant effect on cell proliferation for mutations eliciting substantial decrease in protein stability (Fig. 1c). Only few sporadic mutations (displaying lower significance levels) were able to specifically inhibit ARID1B function (effect on cancer cell proliferation in absence of destabilization), likely due to extensive cooperative binding within the complex impeding the disruption of BRG1 PPI with a single point mutation. Therefore, our DMS data suggest that single point mutations might largely represent loss of function alleles because of an effect on protein stability rather than additional mechanisms (for example, disruption of the protein–protein interaction). To validate our findings, we selected a subset of mutants affecting protein stability as well as cancer cell proliferation (called 'prolif') and a subset affecting protein stability in absence of antiproliferative effect (called 'inert') (boxes in Fig. 1c and Extended Data Fig. 5a). We selected mutants within 10–15 Å distance from BRG1, attempting to balance the type of mutations (Extended Data Fig. 5a). Using the stability sensor assay we validated that all the selected mutants affect protein stability in both human embryonic kidney 293A (HEK293A) and Cal51 cells (Fig. 1d and Extended Data Fig. 5b), as well as we reproduced the pattern of proliferative effect on Cal51 cells using both colony formation assay (Fig. 1e) and live-monitoring cell growth assay (Extended Data Fig. 5c). To explore the mechanism underlying the effect on cancer cell proliferation elicited by 'prolif' mutants compared to the 'inert' ones, we analyzed their potential for complex assembly. Coimmunoprecipitation assays reveal that mutants affecting cancer cell proliferation cannot be assembled into the BAF complex (Fig. 1f), suggesting that exploitation of the synthetic lethal relationship between ARID1B and *ARID1A*[mut] cancers requires profound perturbation of BAF complex composition beyond ARID1B partial protein degradation.

Given the paucity of point mutations affecting cell proliferation without altering protein stability, we extended our analysis by performing DMS screen using our stability sensor screen with the additional four sublibraries encompassing the entire EHD2 domain (Extended Data Fig. 2c). After quality control of the libraries (Extended Data Fig. 3b and Supplementary Table 2), we conducted differential analysis for each individual library (Extended Data Fig. 4b and Supplementary Table 3) and overlaid the results with a set of structural features of the ARID1B EHD2 domain (Fig. 2a and Extended Data Fig. 6a). Overall, we found that positions in the hydrophobic core are more susceptible to mutations than surface-exposed residues (Extended Data Fig. 7a), which is in line with previous studies using DMS coupled to phenotypic readouts[12] or measuring directly epistatic interactions[13] or protein abundance readout[10,14]. Mutations of hydrophobic side chains in helices to charged or polar amino acids caused protein destabilization, while introducing smaller hydrophobic residues (for example, alanine), on average, did not affect stability. Cases where alanine substitutions led

to a marked effect on protein stability were limited to positions with large hydrophobic residues as wild-type amino acid (for example, Phe, Leu, Ile). Proline mutations in helices were particularly destabilizing due to its irregular geometry that disrupts the hydrogen bonding pattern in helices (Extended Data Fig. 7b–h). Within the EHD2 domain, we observed that the central helices were particularly sensitive to mutations, in contrast to unstructured regions and peripheral helices with higher degrees of solvent-exposure (Fig. 2b).

Next, we reasoned that having generated a systematic ARID1B mutation-stability map could shed light on the functional relevance of *ARID1B* clinical missense mutations reported in the ClinVar database. Seven out of eight known pathogenic missense variants were covered by our DMS screen. Six out of these seven showed a strong negative stability score and are located in positions that we found to be highly susceptible to destabilization in general, while only one (E2029K) showed a marginal effect on stability (Fig. 2c). Mutations with uncertain interpretation had varying effects (Fig. 2c). We then essayed the functional impact of a subset of ClinVar mutations on orthogonal readouts of protein stability (Fig. 2e and Extended Data Fig. 8a,b) and cancer cell proliferation (Extended Data Fig. 8c,d), and assessed BAF complex assembly by immunoprecipitation (Extended Data Fig. 8e). Overall, we confirmed that decreased protein levels are the best predictor of CSS pathogenicity, underlining the importance of a tightly regulated ARID1B protein levels during development (as evidenced by the haploinsufficiency of ARID1B mutations in mice[15] and humans[16,17]).

Further, out of the 19 theoretically possible mutations, the clinically manifested mutations in CSS were, in all cases, among the top most destabilizing mutations for the respective position (Extended Data Fig. 9a,b). By contrast, benign CSS mutations and mutations identified in patients without CSS patients (from the GnomAD database) did not impair ARID1B stability (Extended Data Fig. 9c). Structurally, all seven pathogenic mutations are located in the central helices that are particularly critical for ARID1B stability (Fig. 2d). For example, V1939G, M1952T, I2018T and S2019F, cluster in close spatial proximity (<5 Å), supporting that this region is of particular importance to maintain protein integrity. Moreover, the missense variants S2123P and L2070P also showed strong negative stability scores, which is likely caused by perturbing the helical secondary structure (Extended Data Fig. 10a). Thus, we conclude that the pathogenicity of clinically observed mutations in patients with CSS is due to ARID1b destabilization or misfolding, rather than a direct effect on the molecular interaction with another BAF complex subunit.

Together, our DMS data shed light on the functional consequence of *ARID1B* missense mutations by: (1) pinpointing regions and/or positions that are generally critical for ARID1B stability (for example, central helices) and thus are largely intolerant to almost any mutation, and (2)

**Fig. 1 | Point mutations in ARID1B disrupt its function largely by compromising protein stability. a**, Schematic diagrams of *ARID1B* domains (coordinates refer to UniProt variant Q8FND5-3) and distribution of different types of mutation reported from the ClinVar database (https://www.ncbi.nlm.nih.gov/clinvar/, October 2022 release). **b**, Schematic diagram describing the workflow used to perform DMS screens on ARID1B. After library construction and lentiviral transduction, two assays (with their respective libraries) were interrogated, one measuring ARID1B-driven cell proliferation and the other measuring protein stability. **c**, Scatter plot depicting the effect of each mutation in the libraries Pool4 and Pool5 on cell proliferation and protein stability. Dots are colored based on the $-\log_{10}$ adjusted *P* value (adj*P*) for the stability screen and size is based on $-\log_{10}$ (adj*P*) from the proliferation screen. Blue and orange squares represent cutoffs used to select 'inert' and 'prolif' mutants, respectively. **d**, Boxplot representing the percentage of cells gated as GFP<mCherry 'destabilized population' (refer to diagram in **b** and gating scheme in Extended Data Fig. 2b) for the selected set of 'prolif' and 'inert' mutants (Extended Data Fig. 5a) in Cal51 cells. A boxplot represents the median, first and third quartile

and whiskers extend to 95th percentile resulting from six biological replicates. A dotplot underneath each boxplot panel represents values from pooled stability screen colored by $\log_2$ fold change (FC) and sized by $-\log_{10}$ (*P*). Squares underneath the dotplot as well as coloring boxplot coloring represent the category of mutants analyzed. **e**, Boxplot representing colony formation assay data from cell lines stably expressing the indicated ARID1B mutant cDNA. Data are presented as a percentage of surviving cells relative to NoDox control (shRNA to knockdown endogenous ARID1B is doxycycline inducible). Data are depicted as boxplots (representing median, first and third quartile and whiskers extend to 95th percentile) from three biological replicates. Dotplots underneath each boxplot panel represent values from pooled proliferation screen colored by $\log_2$ fold change and sized by $-\log_{10}$ (*P*). Squares underneath the dotplot as well as the boxplot coloring represent the category of mutants analyzed. **f**, Coimmunoprecipitation assay in HEK293A *ARID1A/B* double knockout cells transfected with HA-tagged cDNA constructs indicated. Coloring represent the mutant category (as reported in Fig. 2d,e). Data are representative example from three biological replicates. IP, immunoprecipitation; WT, wild-type.

by providing a comprehensive mutation-stability map to rationalize the interpretation of any known or yet unknown amino acid substitution, including mutations considered 'uncertain' (Extended Data Fig. 10b,c).

In summary, our DMS framework integrates functional and protein stability measurements, which should be broadly applicable to facilitate the interpretation of missense mutations and contribute

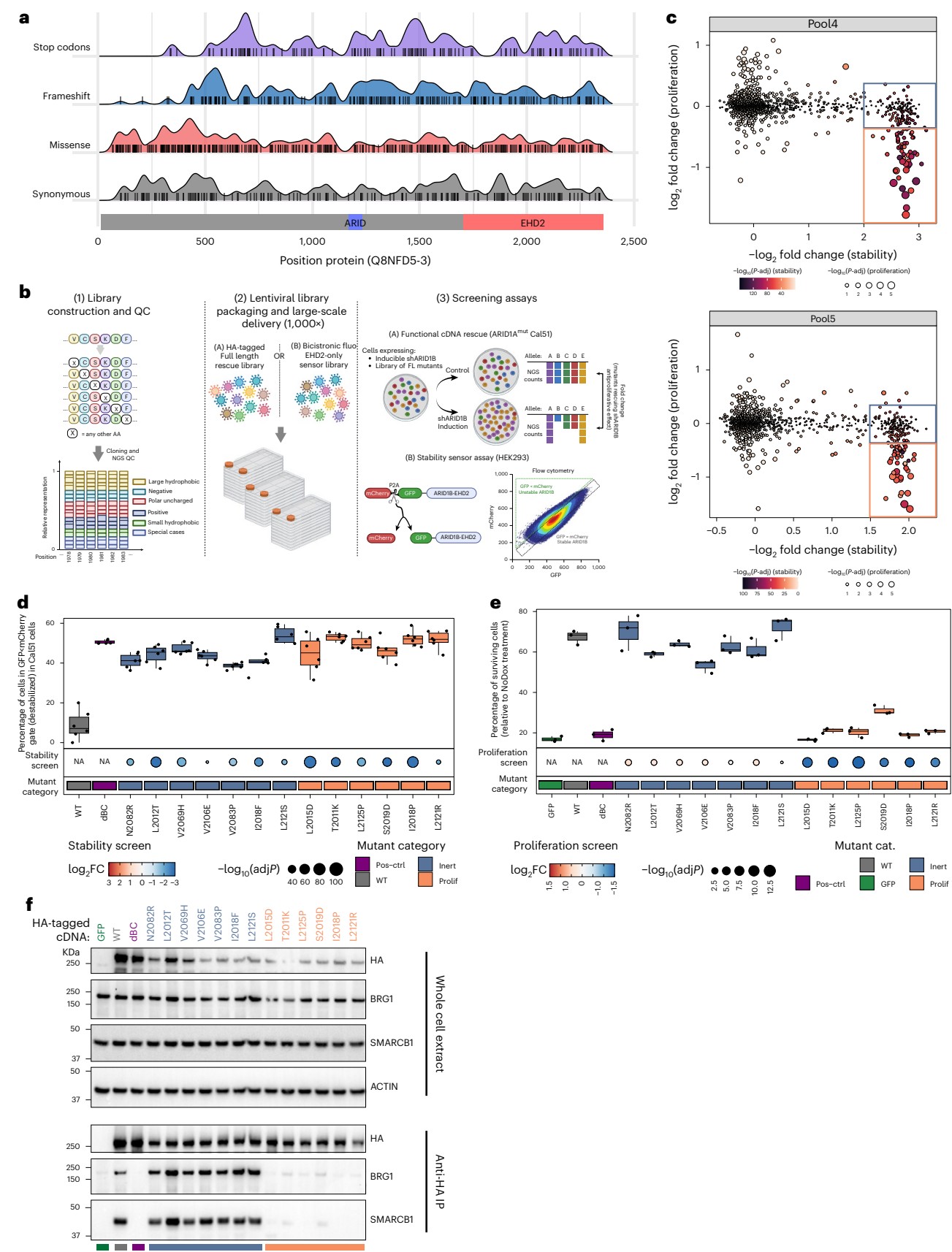

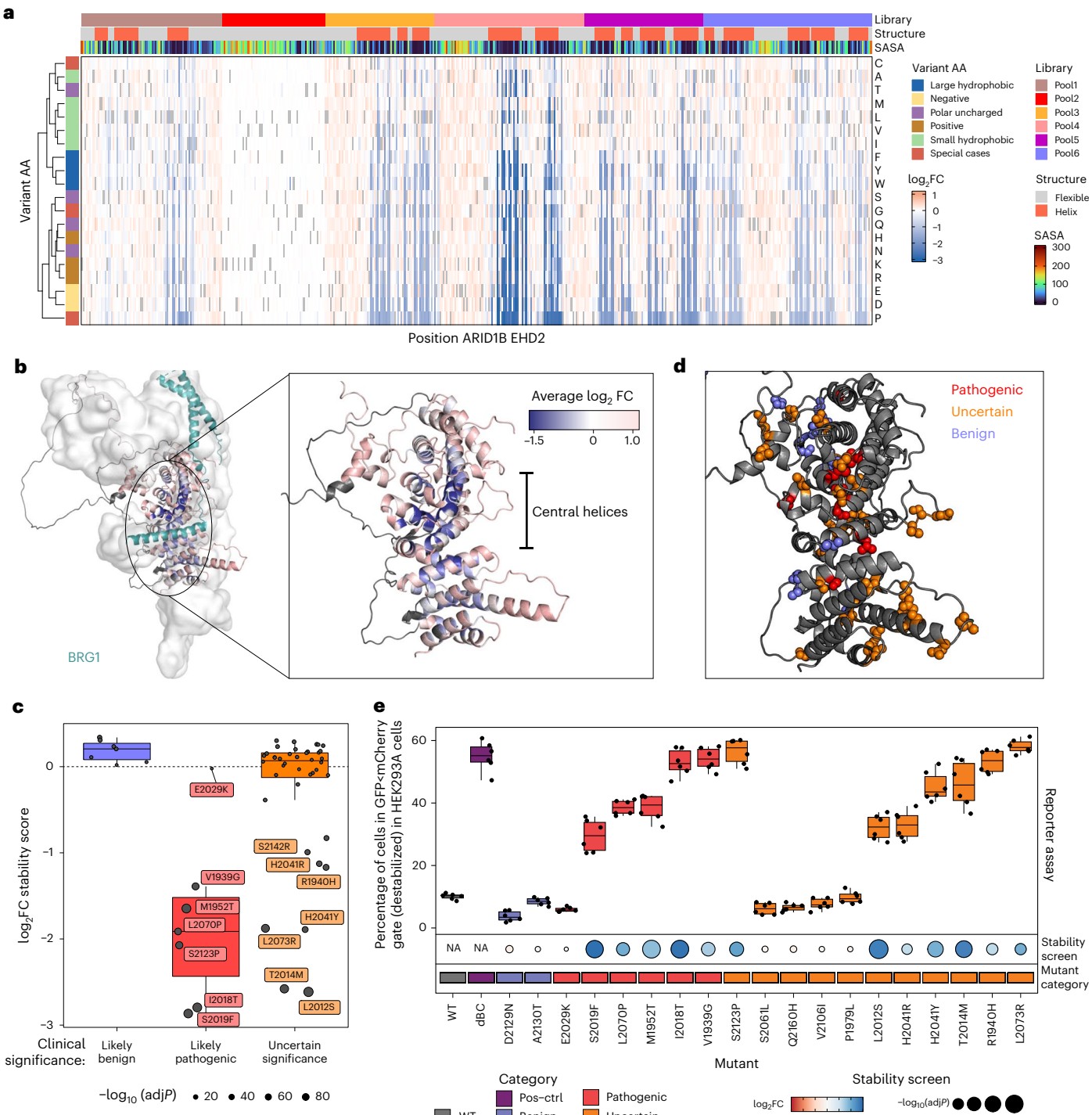

**Fig. 2 | Protein destabilization underlies ARID1B pathogenic missense mutations in CSS. a**, Heatmap depicting protein stability score for each mutation interrogated in ARID1B DMS. Positions coordinates refer to UniProt variant Q8FND5-3 and are annotated based on library pools, presence of a secondary structure (helix) and wild-type amino acid (AA). Variants amino acids are color coded based on the amino acid property. Stability sensor $\log_2$ fold change values are represented in red-white-blue gradient. Gray values represent wild-type amino acids. Solvent accessible surface area (SASA) per residue in Å$^2$ is depicted with rainbow color scale. **b**, ARID1B EHD2 domain alphafold model color coded based on average stability score for each position. BRG1 helix is colored in acquamarine and other BAF complex subunits are depicted as gray surface. The inset shows a magnification of ARID1B EHD2 domain alone. **c**, Boxplot representing the stability score for missense mutations in *ARID1B* EHD2 domain annotated according to clinical evidence of pathogenicity in ClinVar database. Boxplots represent median

and first and third quartiles, and whiskers extend to 95th percentile. **d**, ARID1B EHD2 domain alphafold model in dark gray with positions annotated as mutated in ClinVar displayed as spheres and color coded based on clinical evidence of pathogenicity. BRG1 helix is colored in acquamarine and other BAF complex subunits are depicted as a gray surface. **e**, Boxplot representing the percentage of HEK293A cells gated as GFP<mCherry 'destabilized population' (refer to diagram in **b** and gating scheme in Extended Data Fig. 2b) for a set of mutants reported in ClinVar. As positive control for destabilization ("Pos-ctrl") we used a construct bearing deletion of the BC-Box. The boxplot represents the median, first and third quartile and whiskers extend to 95th percentile resulting from six biological replicates. The dotplot underneath the boxplot panel represents values from pooled stability screen colored by $\log_2$ fold change and sized by $-\log_{10}$ (P). Squares underneath the dotplot as well as the boxplot coloring represent the category of clinical significance reported in ClinVar.

to a deeper understanding of the molecular mechanisms underlying various diseases.

## Online content

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

## Methods

### Cell culture

HEK293A and Cal51 cells were cultured in DMEM (Bioconcept) supplemented with 10% fetal bovine serum (FBS) (Seradigm), 1× L-glutamine (2 mM), 1× sodium pyruvate (1 mM), and 1× nonessential amino acid (0.1 mM). HEK293A cells were obtained from Thermo Scientific and Cal51 from DMSZ, and were tested for identity by single-nucleotide polymorphism genotyping and mycoplasma contamination. The doxycycline-inducible short-hairpin RNA (shRNA) Cal51 cell line was generated by lentiviral transduction of pLKO-TET-ON construct containing the following shRNA sequence: shARID1B_2683 5′-gagagtcacacaaaggaatct-3′. HEK293T *ARID1A/B* double knockout cells were generated by transfecting all-in-one CRISPR plasmids expressing the following single-guide RNA sequences: sgARID1B_2 (5′-ACCGTGAGGTGCCAACGTTTAGGT-3′) sgARID1B_3 (5′-ACCGAAACTTGATAAGCTTCCTAG-3′), sgARID1B_8 (5′-ACCGGGCACCCCACTATACGCTGG-3′), sgARID1A_2 (5′-ACCGTTGAGATGTCCAAACACCCA-3′), sgARID1A_3 (5′-ACCGGAT GTTGGCGAGTGTAACCA-3′) and sgARID1A_4 (5′-ACCGCTTGCAACCAA CCTCAATGT-3′). Cells were then transfected with plasmids expressing ARID1B cDNAs cloned in a custom lentiviral vector under an EF1a promoter.

### Immunoprecipitation and western blotting

For immunoprecipitation assays, cells transfected with various HA-tagged constructs were collected and lysed in RIPA buffer supplemented with protease inhibitor cocktail (Roche). Cleared lysates were incubated with Anti-HA magnetic beads (ThermoFisher catalog no. 88836) overnight at 4 °C. Beads were then washed three times in RIPA buffer and proteins eluted by boiling for 5 min in laemmli buffer. For western blot analyses, cells were collected and lysed in RIPA buffer supplemented with protease inhibitor cocktail (Roche). Protein samples (from cell collection or immunoprecipitation elutions) were loaded on 3–8% Tris-Acetate gels (Invitrogen), transferred onto nitrocellulose membranes and probed with the following antibodies: Actin (Millipore, catalog no. MAB1501; 1:1,000 dilution), HA (Cell Signaling, catalog no. 3724; 1:1,000 dilution), ARID1B (Sigma, catalog no. WH0057492M1, 1:500 dilution), SMARCB1 (Cell Signaling, catalog no. 91735, 1:1,000 dilution), BRG1 (Abcam, catalog no. ab110641, 1:1,000 dilution) and HRP-antirabbit and HRP-antimouse (Cell Signaling, 1:2,500 dilution).

### Cloning of DMS libraries

Six DMS libraries were designed to cover the whole EHD2 domain of ARID1B: pool 1 (1616–1700), pool 2 (1764–1824), pool 3 (1905–1975), pool 4 (1970–2065), pool 5 (2065–2130) and pool 6 (2131–2236). The oligo pools of the DMS libraries were purchased from Twist Bioscience as a Single Site Variant Library with changes at the amino acid level. The oligos were flanked by the sequences 5′-gccatccagaagacttaccgcgtctcg-3′ and 5′-gcagtctggaagacggaaaccgtctcg-3′, which contain BsmbI restriction sites. Oligo pools were amplified by polymerase chain reaction (PCR) using matching primers for the flanking sequences, and cloned into two different backbones under the EF1a promoter by Golden Gate. For the proliferation assay, libraries were cloned into a lentivirus vector containing FL ARID1B while for the sensor assay, libraries were cloned into a bicistronic lentiviral vector expressing mCherry-P2A-eGFP-ARID1B_EHD2. Endura electrocompetent cells (Lucigen) were transformed according to the manufacturer's protocol. We estimated that the transformation efficiency was more than 100-fold over the size of the initial oligo pool, indicating that each variant is highly represented in the plasmid libraries. The bacteria were expanded in Luria-Bertani medium for roughly 16 h (optical density at 600 nm ($OD_{600}$) = 0.8), and plasmid DNA was harvested using a NucleoBond Xtra Maxi kit (Macherey-Nagel). We performed a quality control of all six libraries by NGS. Illumina sequencing libraries were prepared using the NEBNext Ultra II DNA Library Prep Kit and the NEBNext UDI Set 1 (New England Biolabs, catalog nos. E7645L, E6440S), according to the manufacturer's instructions. Paired-end 250 data were produced using the Illumina NovaSeq 6000 system. Quality control analyses (below) retrieved more than 99% of the variants expected in the DMS libraries.

### Proliferation-based assays

For the validation of the proliferation assay, Cal51_shARID1B_2683 cells were transduced with lentivirus plasmids containing either control complementary DNAs (cDNAs) (Nluc or eGFP) or ARID1B constructs under EF1a promoter (multiplicity of infection (MOI) = 0.3). The cells were selected using neomycin (1.2 mg ml⁻¹; Gibco) at 24 h after transduction, after which they were expanded. After 2 weeks, 2,000 cells per well were seeded into six-well plates, in the presence or absence of 100 ng ml⁻¹ doxycycline ($n = 8$ biological replicates) for colony formation assay. At D14, colony formation assays were stopped by adding 3.7% formaldehyde (Sigma), stained using crystal violet (sigma) and quantified. Stable Cal51_shARID1B_2683 cells expressing a battery of ARID1B mutant cDNAs were seeded at 10% confluency in 96-well plates. After 24 h, half of the wells were treated with 100 ng ml⁻¹ doxycycline ($n = 4$ biological replicates) and imaged every 12 h for 15 days for live imaging using an Incucyte SX5 (Sartorius).

For the libraries' screening, Cal51_shARID1B_2683 cells were transduced with independent lentiviral pools (MOI = 0.3) of pool 4 (1970–2065) and pool 5 (2060–2130) libraries. Roughly 1,000 cells per plasmid were transduced to ensure a correct representation of all variants in the cell population. The cells were selected using neomycin (1.2 mg ml⁻¹; Gibco) at 24 h after transduction, after which they were expanded. After 2 weeks, 1.4 million cells were seeded in cell stacks in the presence or absence of 100 ng ml⁻¹ doxycycline ($n = 5$ biological replicates for each condition) to induce ARID1B KD. After 10 days, cells were collected. Genomic DNA (gDNA) was extracted using Dneasy Blood & Tissue Kit (Qiagen), libraries were amplified by PCR and amplicons were purified using SPRI beads (Beckman) before being submitted to NGS. Illumina sequencing libraries were prepared using the NEBNext Ultra II DNA Library Prep Kit and the NEBNext UDI Set 1 (New England Biolabs, catalog nos. E7645L, E6440S), according to the manufacturer's instructions. Paired-end 250 data were produced using the Illumina NovaSeq 6000 system.

### Stability sensor assays

For the validation of the sensor assay HEK293A cells were transfected with 1 μg of pXP1510-mCherry-eGFP-ARID1B (1565–2236) plasmid and derivatives carrying a battery of mutations. X-tremeGENE9 (Roche) was used for the transfection, according to manufacturer's protocol. After 72 h of transfection, mCherry and/or eGFP expression was analyzed by fluorescence activated cell sorting (FACS) (Cytoflex, Beckman Coulter) using CytExpert v.2.4.0.28 software.

For the libraries' screening, HEK293A cells were transduced with independent lentiviral pools (MOI = 0.3) of the six libraries pools ($n = 3$ biological replicates). Roughly 1,000 cells per plasmid were transduced to ensure a correct representation of all variants in the cell population. Cells were expanded and collected at 14 days before cell sorting by FACS sorting (below). Sorted cells were centrifuged, lysed and gDNA was extracted using Dneasy Blood & Tissue Kit (Qiagen). Libraries were amplified by PCR and amplicons were purified using SPRI beads (Beckman) before being submitted to NGS. Illumina sequencing libraries were prepared using the NEBNext Ultra II DNA Library Prep Kit and the NEBNext UDI Set 1 (New England Biolabs, catalog nos. E7645L, E6440S), according to the manufacturer's instructions. Paired-end 250 data were produced using the Illumina NovaSeq 6000 system using Illumina NovaSeq control software v.1.8.1.

### FACS

A FACS flow cytometer (Aria Fusion, Becton Dickinson, equipped with BD FACSDiva Software) was used for cell sorting, using the 70 μm nozzle

at 70 psi pressure, using 1× BioSure Preservative-Free Sheath Solution (Concentrate, catalog no. 1027). Temperature of the sample and of the collecting tubes was set at 4 °C and cells were sorted using a four-way purity mode at an event rate around 11,000 cells per s to maintain a high recovery yield. Cells were sorted into 1.5 ml Eppendorf tubes containing 350 μl of FACS buffer or a 5 ml PP FACS tube containing 350 μl of FACS buffer (DPBS, 2 mM EDTA, 2% FBS).

To match the calculated minimum number of sorted cells with regards to the library structure and sequencing depth, the samples were sorted individually in parallel on different instruments. To avoid batch effects, each instrument's performance was normalized using CST beads' brightest peak signal for all instruments, and an equal fraction of each individual sample was sorted on each instrument in parallel before being pooled.

### Computational analyses
For each sample, paired-end sequencing reads (Fastq) were stitched into unique sequence fragments using NGmerge (v.0.3)[18], stitched reads were mapped to the codon-optimized wild-type reference sequence using bowtie2 (v.2.4.4, ref. 19) and PCR amplification primers were trimmed off using cutadapt (v.3.5, ref. 20). A summary count matrix for each variant (Supplementary Table 3, DMS_data_all and Supplementary Table 2, Plasmid_libraries_QC) was computed using a custom Python and R script (Python v.3.9.9, R v.4.1.10). Differential representation analysis was then performed using DESeq2 (v.1.34.0)[21] to identify variants significantly affecting proliferation or protein stability (Supplementary Table 3, DMS_data_all). All statistical analysis and plotting were performed in R (v.4.1.1). The ARID1B structural model was downloaded from the AlphaFold Protein Structure Database in December 2022, and trimmed to residues 1616–2236. The ARID1B model was aligned to the ARID1A cryo-EM structure accessible under Protein Data Bank code 6LTH, with a root mean-squared deviation of 0.6 Å. The solvent accessible surface area was computed the ARID1B (1616–2236) AF2 model using the Shrake–Rupley 'rolling ball' algorithm, with a probe radius of 1.4 Å. The surface area was assigned on the residue level. Secondary structure assignment for each residue was computed using Pymol v.2.5.2. DMS values were mapped onto the ARID1B model using a custom Python script, and images were rendered using Pymol v.2.5.2. Figure 1b and Extended Data Fig. 3a have been created with BioRender.com.

### Reporting summary
Further information on research design is available in the Nature Portfolio Reporting Summary linked to this article.

## Data availability
All the data have been uploaded to the Sequence Read Archive with the BioProject ID PRJNA1010676 and can be publicly accessed at ID 1010676, BioProject, NCBI (nih.gov). All processed data can additionally be found as Supplementary Tables and have been deposited in Zenodo under the record no. 10418664. Source data are provided with this paper.

## Code availability
Code have been deposited in Github https://github.com/Novartis/dms-pipeline/tree/main and Zenodo under record 10418664.

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

## Acknowledgements
We are grateful to E. Angevaare, T. le Meur, A. Hodzic, C. Walter and N. Duerig for additional support with FACS experiments. We are indebted to C. Velez-Vega and B. Reimer for in silico mutagenesis analyses and F. Gypas for implementation of DMS analysis pipeline. We additionally thank N. Mashtalir, F. Meili, M. Iurlaro and the laboratory of D. Schübeler for discussion.

## Author contributions
F.M.-M., B.B.-P., C.A., M.B., K.C., J.K. and M.A. performed experimental work. S.M., K.S. and F.S. performed computational analyses. U.N. and L.B. supervised genomics and computational work. F.S. and G.G.G. supervised the project and wrote the manuscript.

## Competing interests
All authors are employees and/or shareholders of Novartis Pharma.

## Additional information
**Extended data** is available for this paper at https://doi.org/10.1038/s41594-024-01229-2.

**Correspondence and requests for materials** should be addressed to Fabian Sesterhenn or Giorgio G. Galli.

**Peer review information** *Nature Structural & Molecular Biology* thanks Rugang Zhang and the other, anonymous, reviewer(s) for their contribution to the peer review of this work. Peer reviewer reports are available. Dimitris Typas was the primary editor on this article and managed its editorial process and peer review in collaboration with the rest of the editorial team. Peer reviewer reports are available.

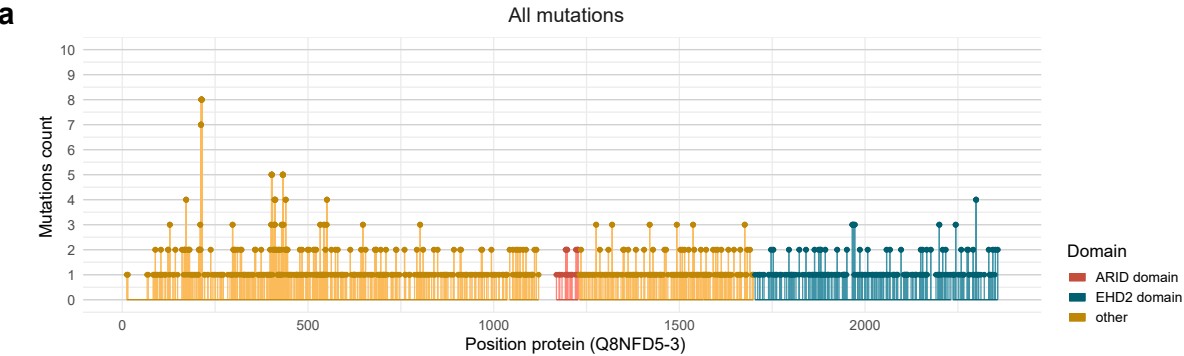

**Extended Data Fig. 1 | Distribution and type of ARID1B mutations in ClinVar database. a**) Lollipop plot representing the number of mutations for each position of *ARID1B* reported in ClinVar database (mapped to *ARID1B* uniprot variant Q8NFD5-3). Positions are colored according to reported positions of ARID or EHD2 domain (BAF250C/DUF3518). **b**) same lollipop plot as in a) but mutations are split in different graphs according to mutation type.

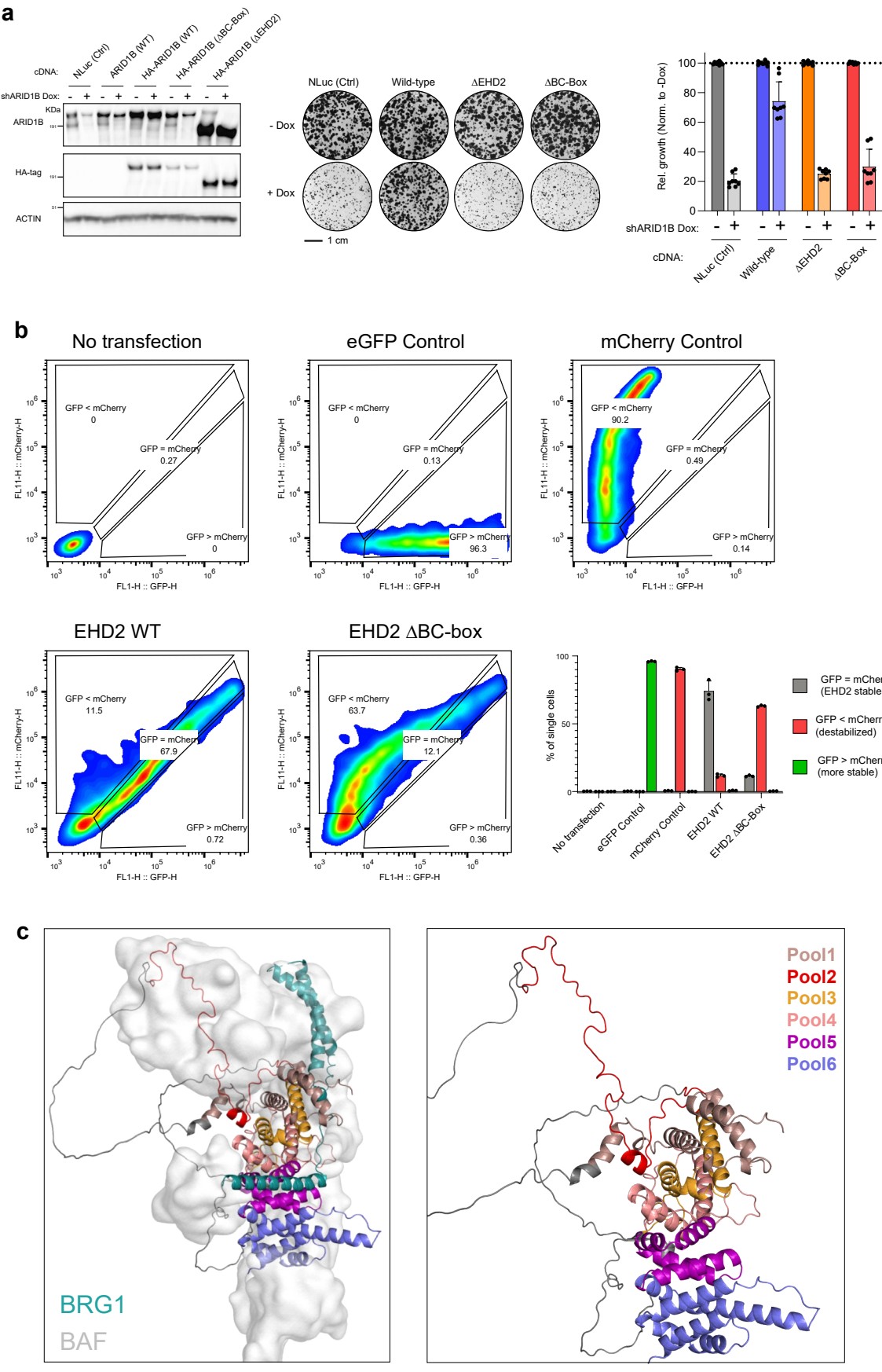

**Extended Data Fig. 2 | See next page for caption.**

**Extended Data Fig. 2 | Setup of cellular assays and library pools for ARID1B DMS screens. a**) Left panel depicts western blot analysis of Cal51 cells bearing doxycycline inducible shRNA against ARID1B and overexpressing different HA-tagged constructs. Actin is used as loading control. Central panel displays a representative colony formation assay at 14 days post-treatment of cells from the same cells. Right panel depicts a histogram plot summarizing the colony formation assay data. Data are presented as mean values +/- SD from 8 biological replicates. **b**) Representative flow cytometry density plots representing distribution of cells according to GFP (x-axis) and mCherry (y-axis) signal. Neighboring histogram plots summarizes the quantification of percentage of cells in each gate. Data are presented as mean values +/- SD from three biological replicates. **c**) ARID1B EHD2 domain alphafold model color coded according to their representation in the different DMS library pools. BRG1 helix is colored in aquamarine and other BAF complex subunits are depicted as transparent grey surface.

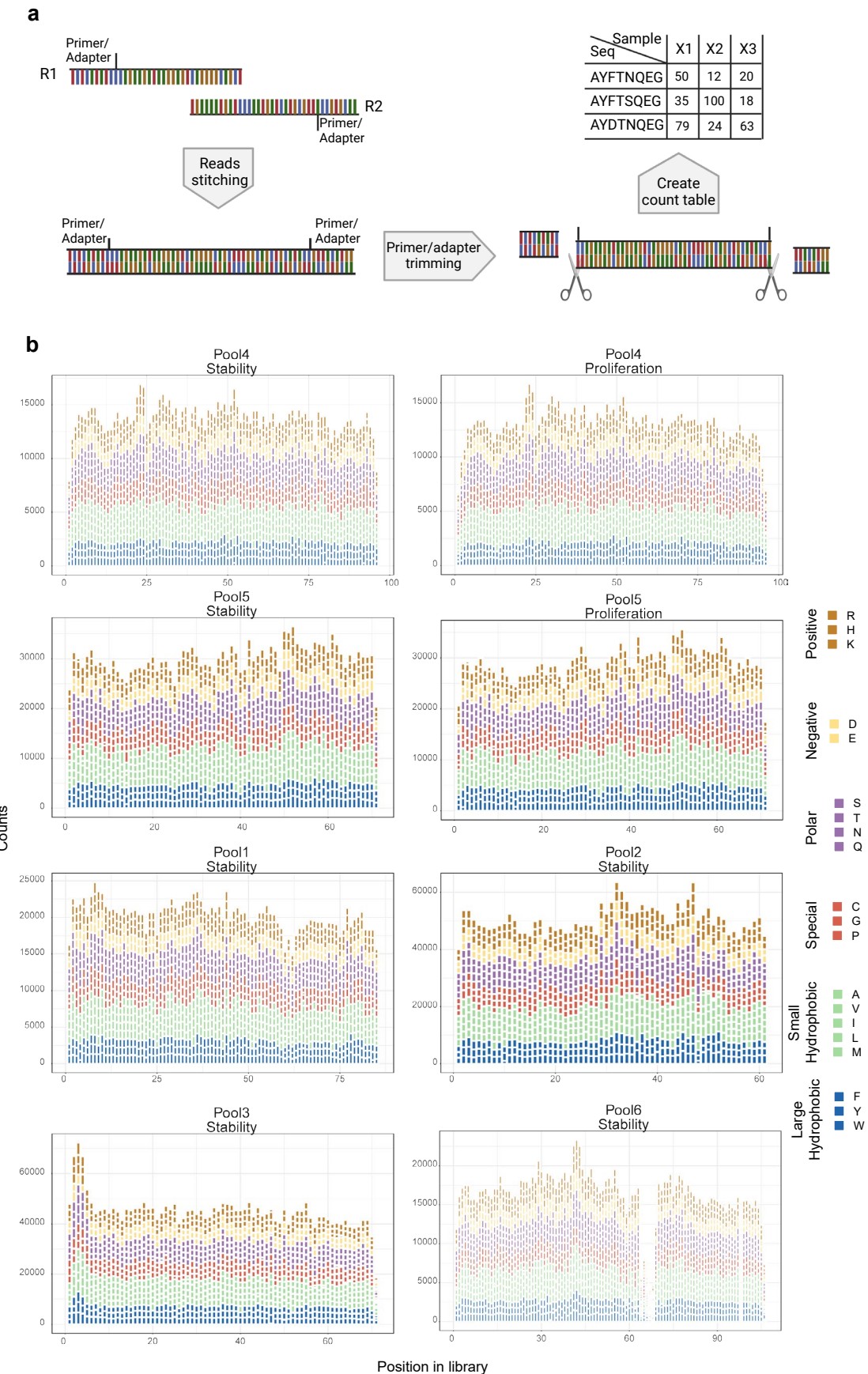

**Extended Data Fig. 3 | Data analysis workflow and library pools QC. a**) Scheme representing the workflow employed for the analysis of DMS NGS data. **b**) Stacked bar plot representing the counts for each variant allele in each DMS Pool cloned. Aminoacids are color coded based on their properties.

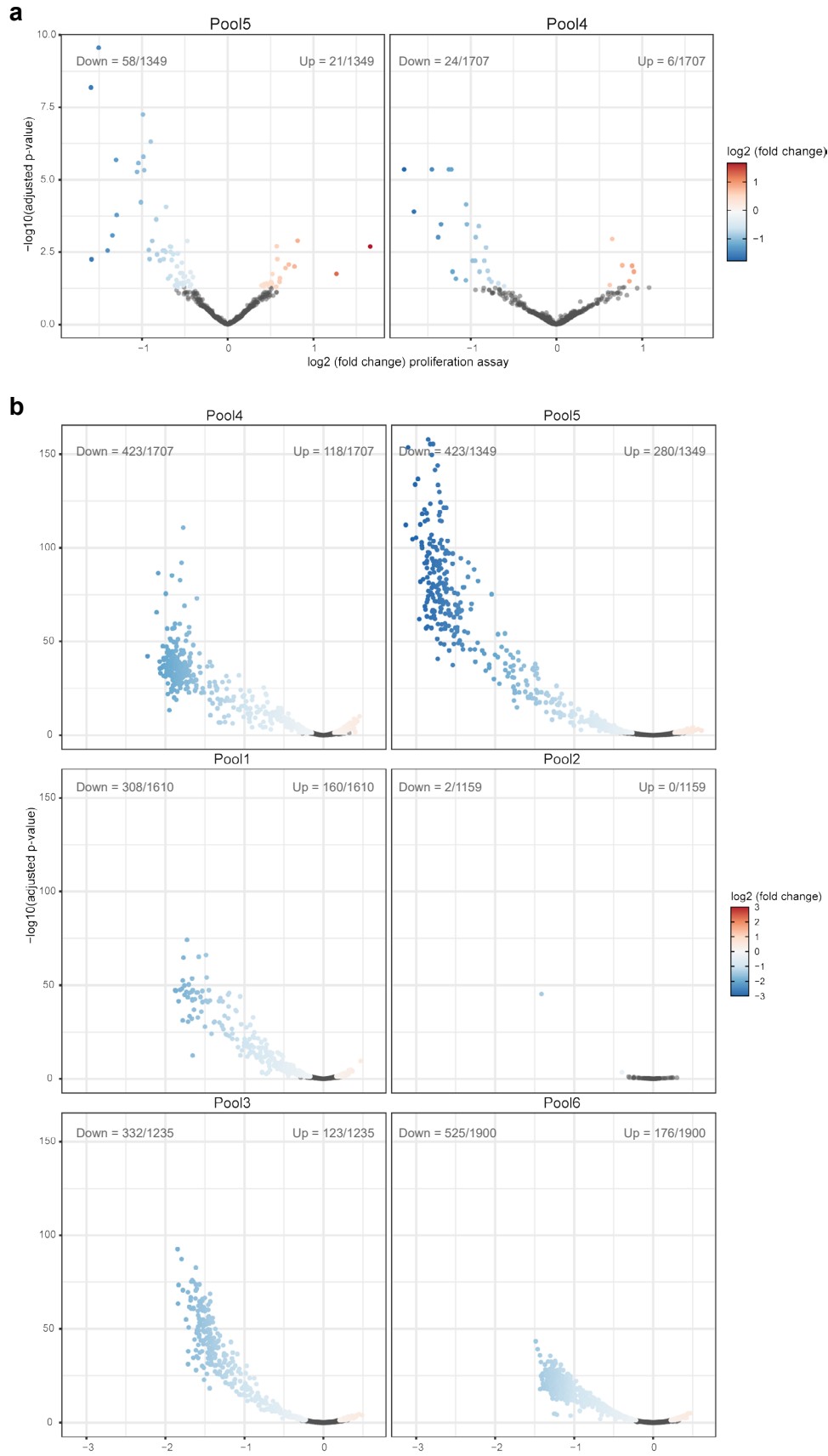

**Extended Data Fig. 4 | See next page for caption.**

**Extended Data Fig. 4 | Differential representation analysis for all ARID1B DMS data. a**) Volcano plot representing differential representation of variants in Pool4 and Pool5 in proliferation assay between untreated and doxycycline-treated (knockdown of endogenous ARID1B via shRNA) cells. x-axis represent log2 fold-change and y-axis -log10 adjusted p-value. Colored dots (red-white-blue scale) represent statistically significant hits. Numbers in top left and right corner indicate numbers of significant hits (adj-pvalue < 0.05) relative to total number of variants in each library. **b**) Volcano plot representing differential representation of variants in each of the 6 pools in stability sensor assay between cells falling in GFP$^{low}$/mCherry$^{high}$ and cells expressing equal levels of GFP and mCherry. x-axis represent log2 fold-change and y-axis -log10 adjusted p-value. Colored dots (red-white-blue scale) represent statistically significant hits. Numbers in top left and right corner indicate numbers of significant hits (adj-pvalue < 0.05) relative to total number of variants in each library.

a



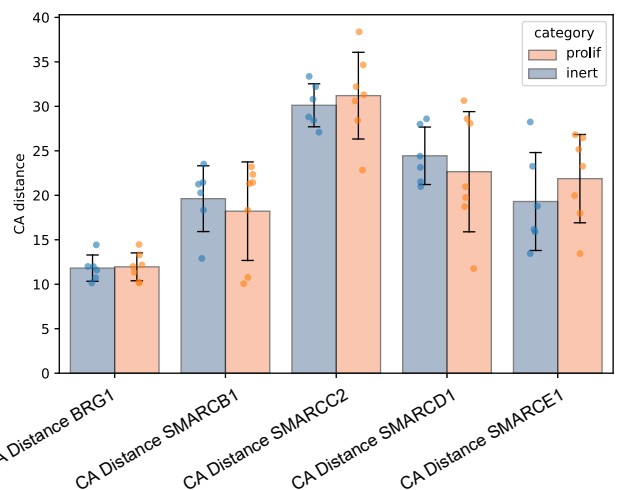

b

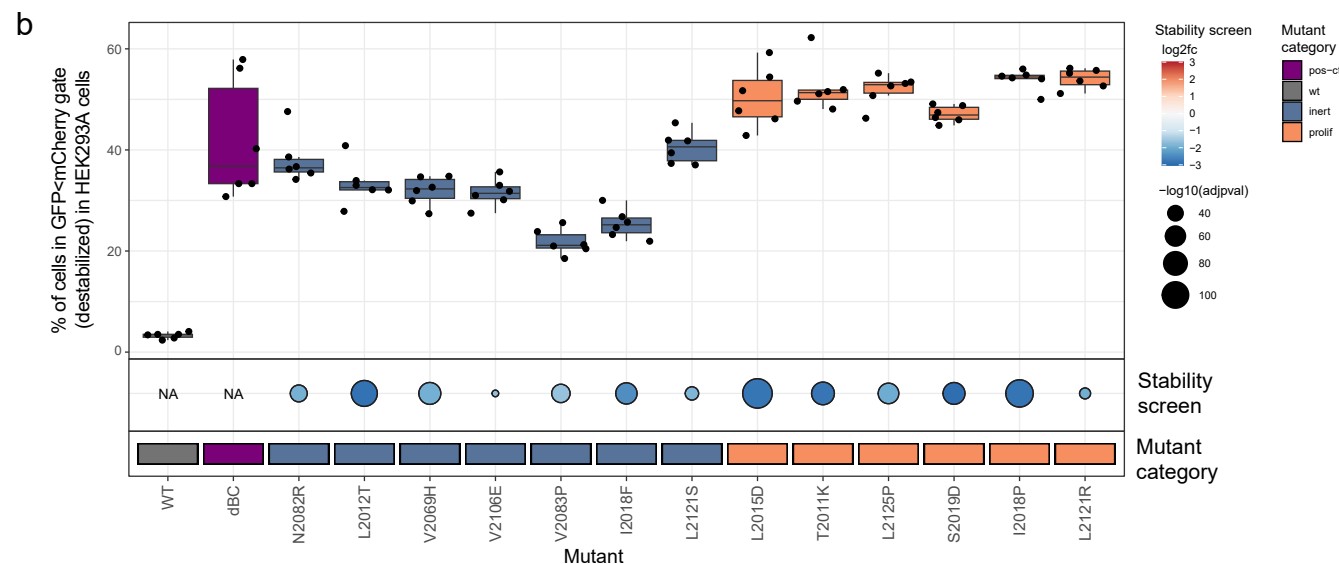

c

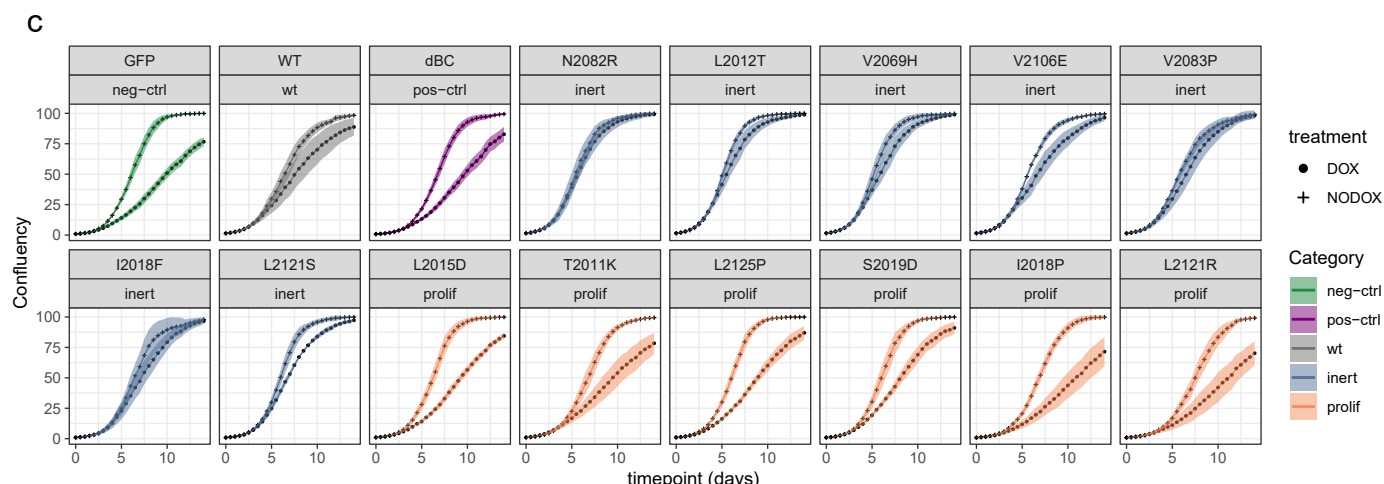

**Extended Data Fig. 5 | See next page for caption.**

**Extended Data Fig. 5 | Selection and validation of 'prolif' and 'inert' mutants.**
**a**) Left panel: list of 'Inert' and 'Prolif' mutants selected using cutoffs reported in Fig. 1c. Right panel: distribution of distances (in Å) from closest residue in the indicated proteins. Data are presented as mean +/- SD. **b**) Boxplot representing the % of cells gated as GFP<mCherry 'destabilized population' (refer to diagram in Fig. 1b and gating scheme in Extended Data Fig. 2b) for the selected set of 'prolif' and 'inert' mutants (see Extended Data Fig. 5a) in HEK293A cells. Boxplot represents median, first and third quartile and whiskers extend to 95th percentile resulting from 6 biological replicates. Dotplot underneath boxplot panel represents values from pooled stability screen colored by log2 fold-change and sized by -log10 (pvalue). Squares underneath the dotplot as well as coloring boxplot coloring represent the category of mutants analyzed. **c**) Growth curve assay measured by live imaging (imaged every 12 hours) for cells expressing different ARID1B mutant cDNAs. 'Plus' and 'filled circle' represent mean confluency and ribbon represent standard deviation of 4 independent replicates. Coloring represent the category of the analyzed mutant.

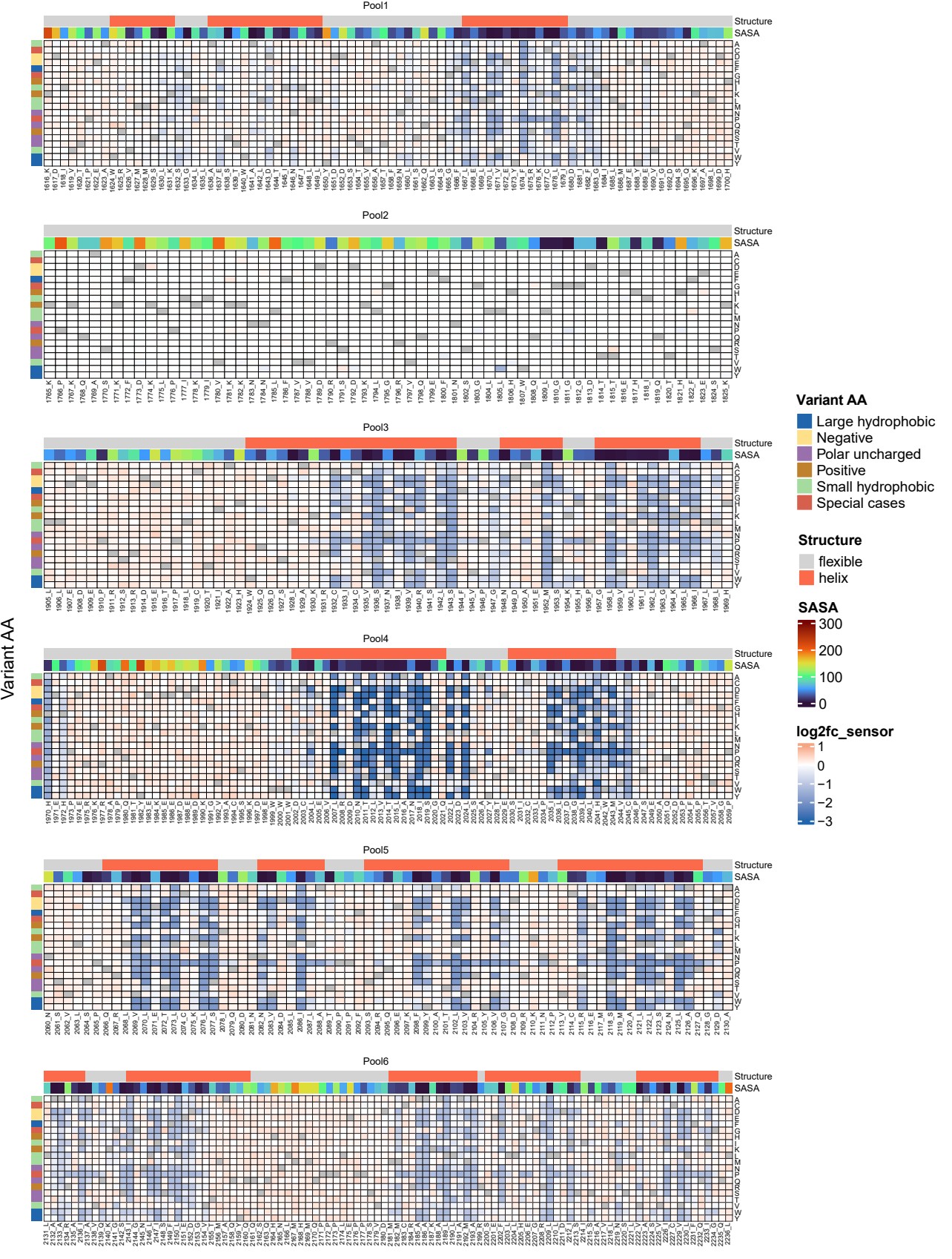

**Extended Data Fig. 6 | See next page for caption.**

**Extended Data Fig. 6 | Results of stability sensor DMS screens for each library pool.** Heatmap depicting protein stability score for each mutation interrogated in ARID1B DMS subdivided by library pool. Positions are annotated based on presence of a secondary structure (helix) and wild type amino acid. Variants amino acids are color-coded based on the amino acid property. Stability sensor log2-fold change values are represented in red-white-blue gradient. Grey values represent wild type aminoacids. Solvent accessible surface area (SASA) per residue in $Å^2$ is depicted with rainbow color scale.

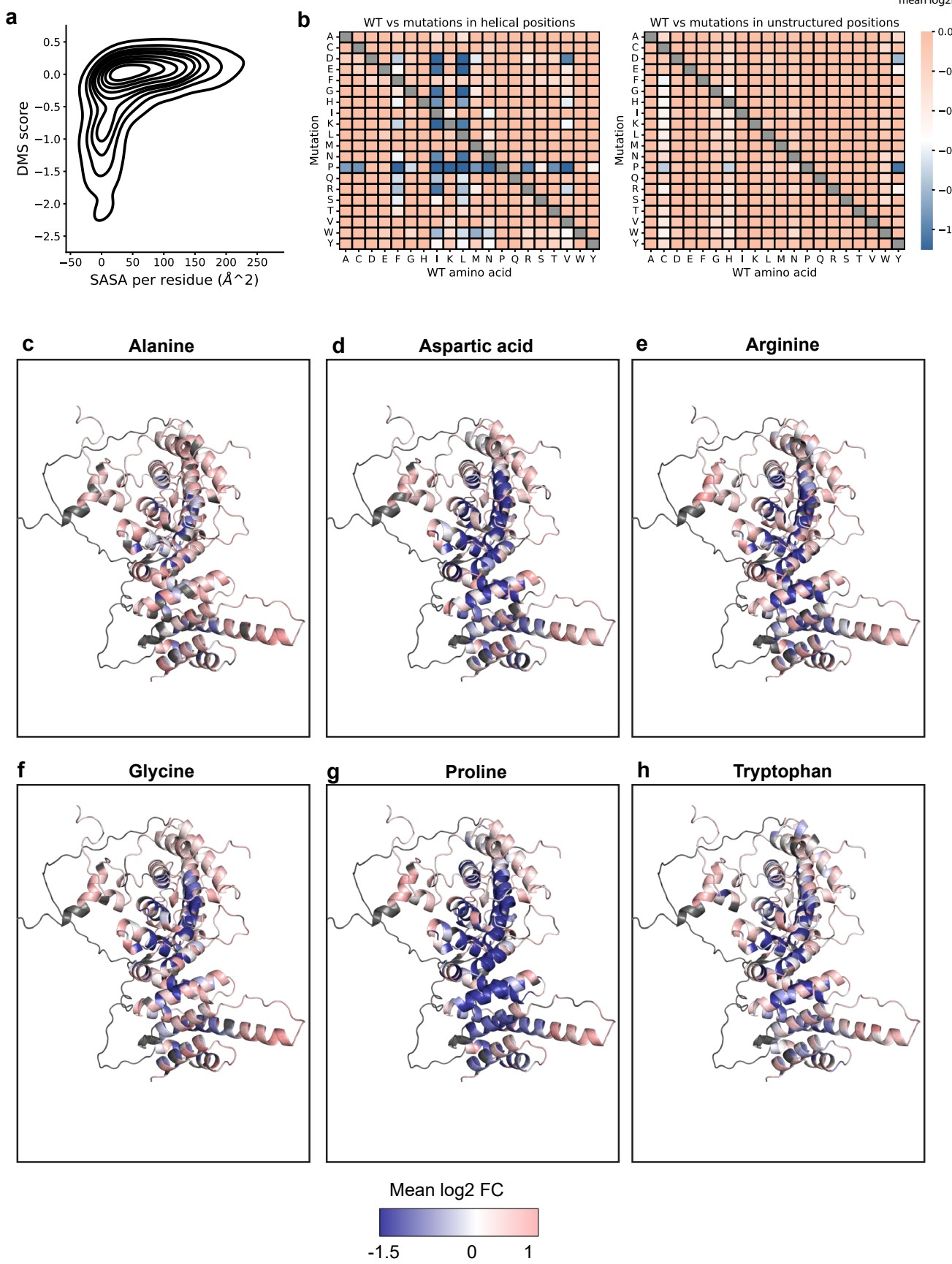

**Extended Data Fig. 7 | See next page for caption.**

**Extended Data Fig. 7 | Relationship between DMS stability score and structural features of ARID1B EHD2 domain and effect of specific substitution on ARID1B EHD2 stability. a)** Density plot showing DMS scores versus solvent accessible surface area (SASA) per residue in Å², computed on the AlphaFold2 structural model of the ARID1B EHD2 domain. **b)** Mean DMS log2FC for WT vs mutated amino acid, averaged over all positions in helices (left) and unstructured regions (right). **c–h)** Structural visualization of mean log2FC values for alanine (c), aspartic acid (d), arginine (e), glycine (f), proline (g), and tryptophane (h) mutations.

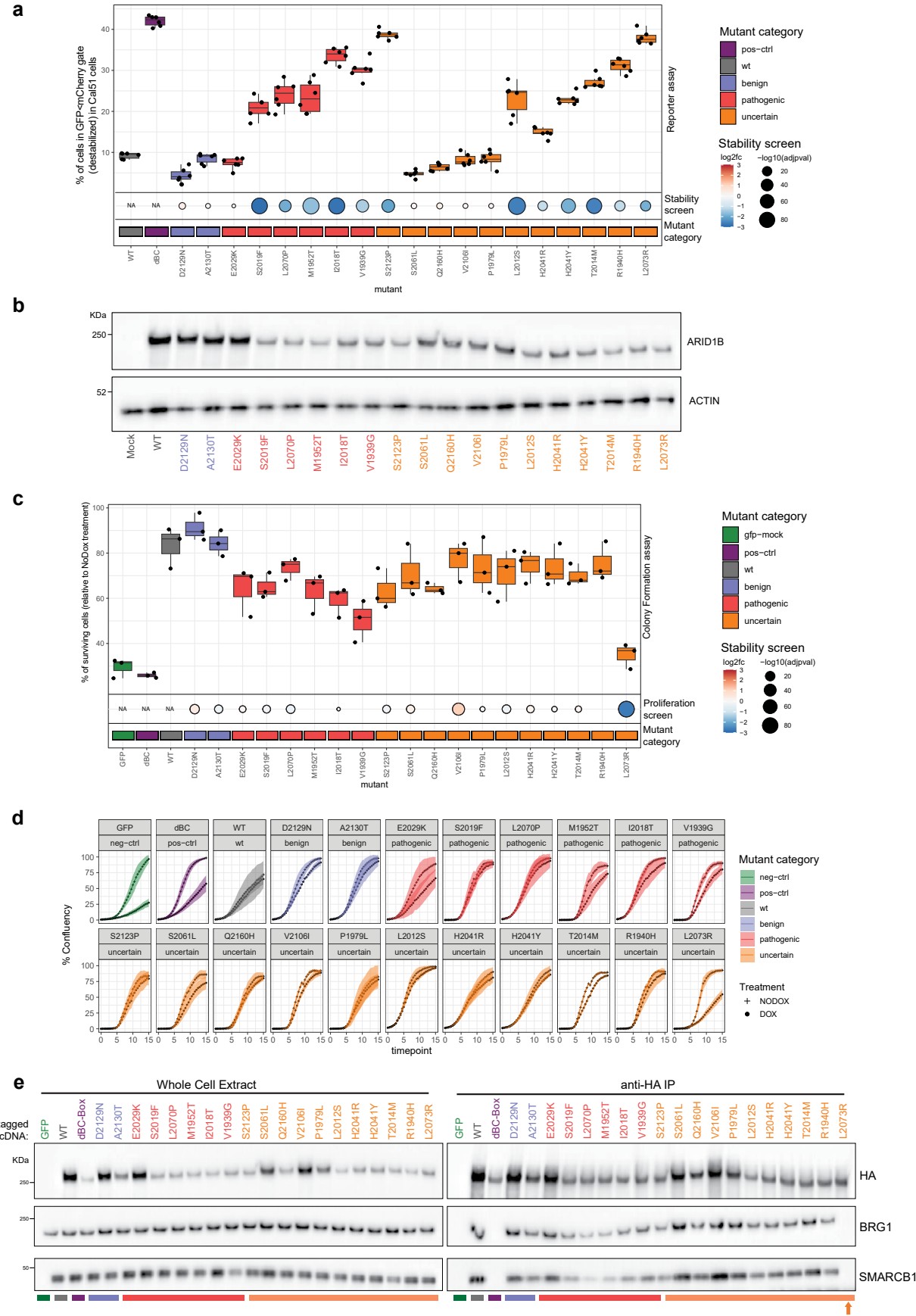

**Extended Data Fig. 8 | See next page for caption.**

**Extended Data Fig. 8 | Validation of functional impact of ARID1B mutation in orthogonal destabilization and proliferation assays. a**) Boxplot representing the % of Cal51 cells gated as GFP<mCherry 'destabilized population' (refer to diagram in Fig. 1b and gating scheme in Figure S2b) for a set of mutants reported in ClinVar. Boxplot represents median, first and third quartile and whiskers extend to 95th percentile resulting from 6 biological replicates. Dotplot underneath boxplot panel represents values from pooled stability screen colored by log2 fold-change and sized by -log10 (pvalue). Squares underneath the dotplot as well as coloring boxplot coloring represent the category of clinical significance reported in ClinVar. **b**) Western blot analysis of mutants reported in a) transfected in HEK293T *ARID1A/B* double KO cells Coloring represent the category of clinical significance reported in ClinVar **c**) Boxplot representing colony formation assay data from cell lines stably expressing the indicated ARID1B mutant cDNA. Data are presented as % of surviving cells relative to NoDox control (shRNA to knockdown endogenous ARID1B is doxycycline inducible).

Data are depicted as boxplot (representing median, first and third quartile and whiskers extend to 95th percentile) from three biological replicates. Dotplot underneath boxplot panel represents values from pooled proliferation screen colored by log2 fold-change and sized by -log10 (pvalue). Squares underneath the dotplot as well as coloring boxplot coloring represent the category of clinical significance reported in ClinVar. **d**) Growth curve assay measured by live imaging (imaged every 12 hours) for cells expressing different ARID1B mutant cDNAs. 'Plus' and 'filled circle' represent mean confluency and ribbon represent standard deviation of 4 independent replicates. Coloring represent the category of clinical significance reported in ClinVar. **e**). Co-immunoprecipitation assay in HEK293A *ARID1A/B* double KO cells transfected with HA-tagged cDNA constructs indicated. Coloring represent the mutant category (as reported throughout the figure). Orange arrow indicate that only L2073R mutant fails to be incorporated into BAF complex.

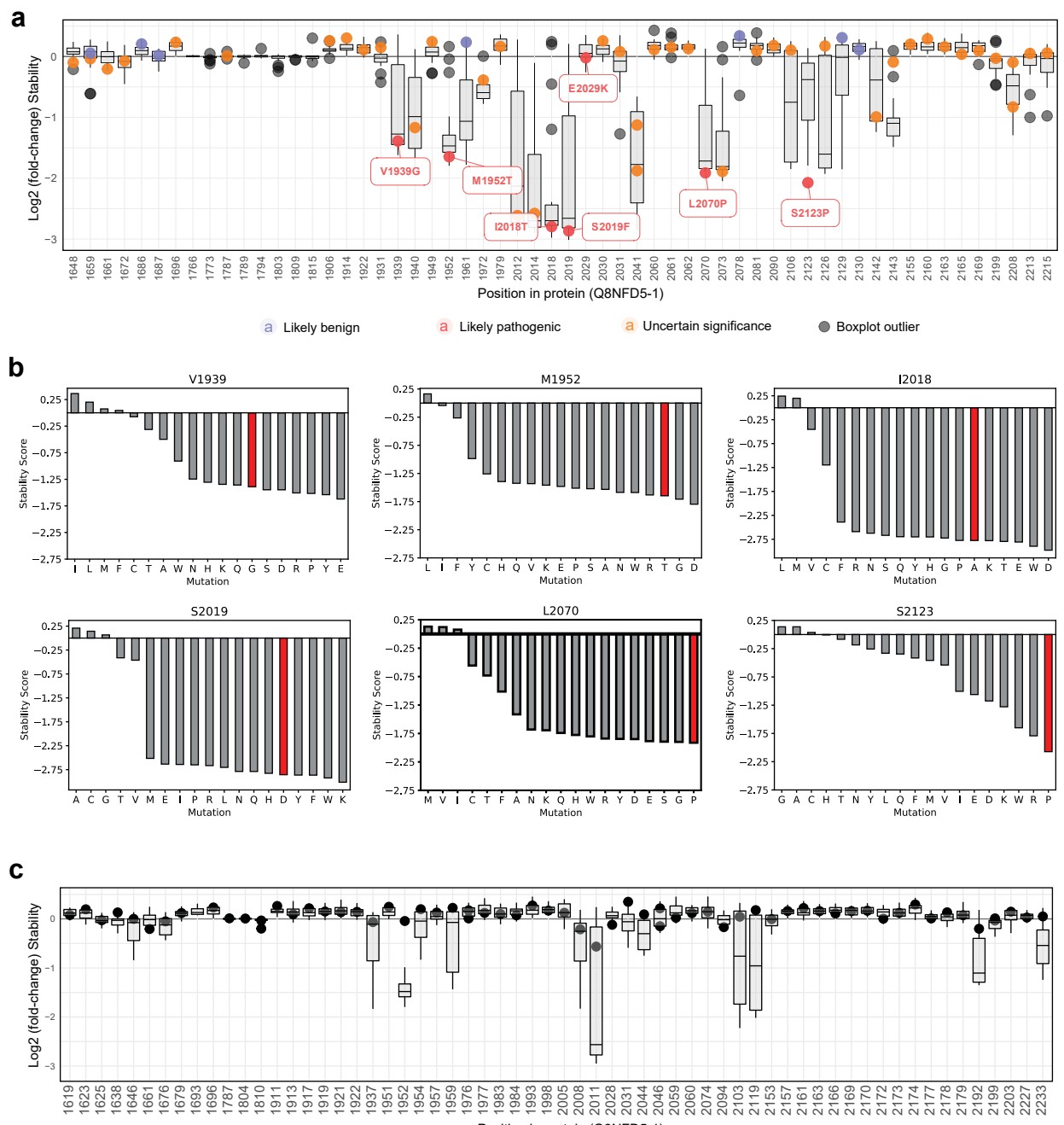

**Extended Data Fig. 9 | Effect of specific codon changes in ARID1B Missense mutations. a)** Boxplot of distribution of stability score for all 19 aminoacid substitutions in positions reported to bear missense mutations in ClinVar. Points represent either outliers from the distribution (in grey) or the actual substitution detected in the patient (colored according to pathogenicity). Boxplots represent median and first and third quartiles, and whiskers extend to 95th percentile. **b)** DMS stability score (log2FC) for six positions that have pathogenic missense mutations annotated in ClinVar. Known pathogenic mutations are highlighted in red, among all other mutations tested in DMS (grey). **c)** Similar plot as in a) but representing mutations reported in non-CSS patients as retrieved from GnomAD database. Boxplots represent median and first and third quartiles, and whiskers extend to 95th percentile.

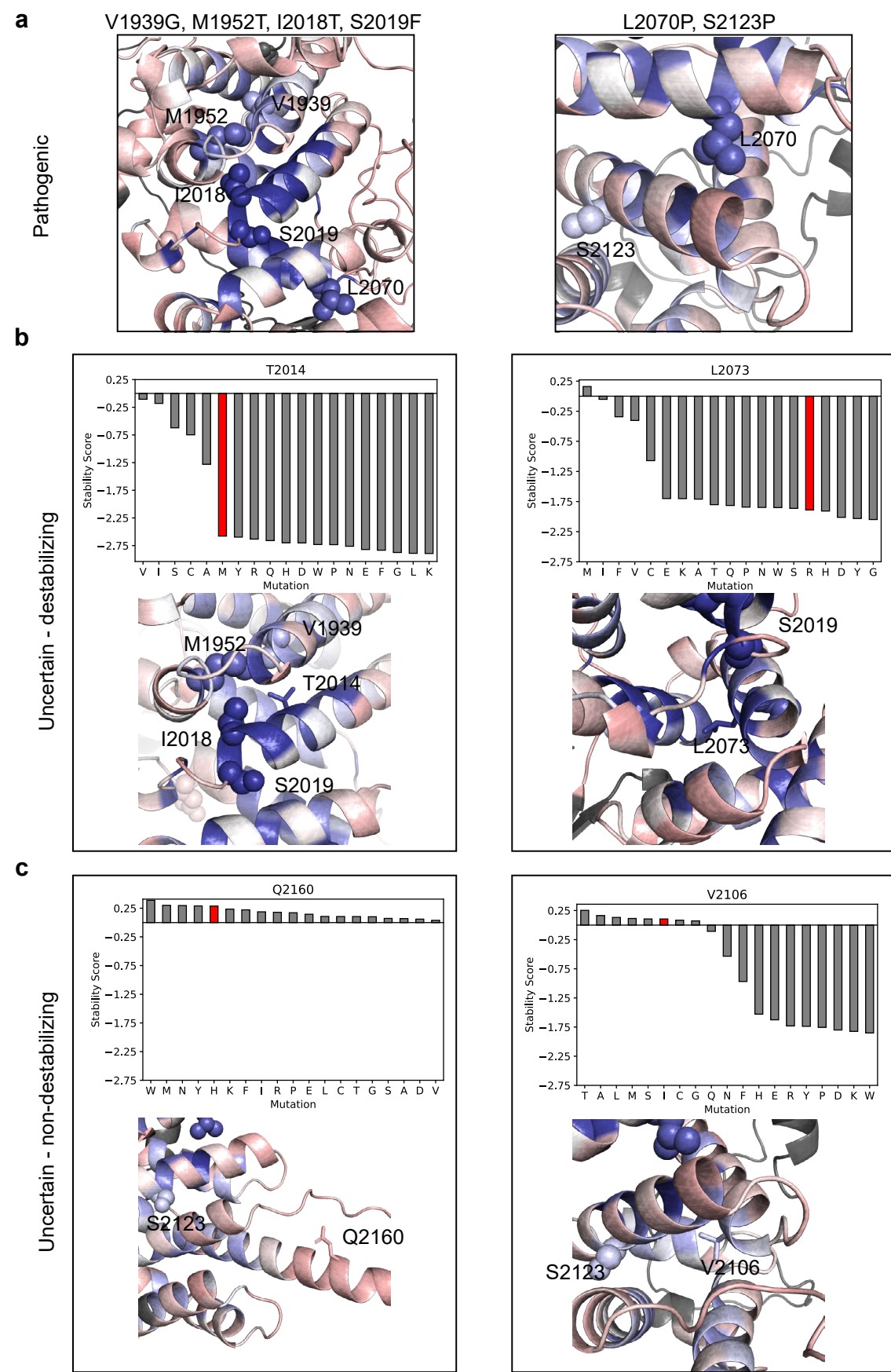

**Extended Data Fig. 10 | See next page for caption.**

**Extended Data Fig. 10 | Evaluation of ARID1B mutations with different pathogenic potential. a**) Close up view on pathogenic mutants V1939G, M1952T, I2018T, S2019F, L2070P and S2123P. **b**) Uncertain mutations T2014M and L2073R. DMS stability data and structural location indicate that these mutations likely destabilize ARID1B, similarly to known pathogenic mutations. **c**) Uncertain mutations Q2160H and V2106I, for which DMS data show no effect on ARID1B stability. Pathogenic mutations are shown as spheres, uncertain mutations are shown as stick representation and colored according to the mean DMS stability score at the respective position (log2FC).

# Reporting Summary

## Statistics

For all statistical analyses, confirm that the following items are present in the figure legend, table legend, main text, or Methods section.

| n/a | Confirmed | |
|---|---|---|
| ☐ | ☒ | The exact sample size ($n$) for each experimental group/condition, given as a discrete number and unit of measurement |
| ☐ | ☒ | A statement on whether measurements were taken from distinct samples or whether the same sample was measured repeatedly |
| ☐ | ☒ | The statistical test(s) used AND whether they are one- or two-sided <br> *Only common tests should be described solely by name; describe more complex techniques in the Methods section.* |
| ☒ | ☐ | A description of all covariates tested |
| ☐ | ☒ | A description of any assumptions or corrections, such as tests of normality and adjustment for multiple comparisons |
| ☐ | ☒ | A full description of the statistical parameters including central tendency (e.g. means) or other basic estimates (e.g. regression coefficient) AND variation (e.g. standard deviation) or associated estimates of uncertainty (e.g. confidence intervals) |
| ☐ | ☒ | For null hypothesis testing, the test statistic (e.g. $F$, $t$, $r$) with confidence intervals, effect sizes, degrees of freedom and $P$ value noted <br> *Give P values as exact values whenever suitable.* |
| ☒ | ☐ | For Bayesian analysis, information on the choice of priors and Markov chain Monte Carlo settings |
| ☒ | ☐ | For hierarchical and complex designs, identification of the appropriate level for tests and full reporting of outcomes |
| ☒ | ☐ | Estimates of effect sizes (e.g. Cohen's $d$, Pearson's $r$), indicating how they were calculated |

*Our web collection on statistics for biologists contains articles on many of the points above.*

## Software and code

Policy information about availability of computer code

| Data collection | Standard software from manufacturers (Illumina Inc., Illumina NovaSeq control software Version 1.8.1 ) for NGS-based data collection has been used. Additionally basecalling was performing with RTA v3.4.4. Post-run processing was performed with BCL2FASTQ v2.20.0.422. Run QC was performed with FASTQC v0.11.9. <br> For Flow Cytometry based assay: CytExpert 2.4.0.28 for analysis and BD FACSDiva™ Software v9.0 for FACS sorting |
|---|---|
| Data analysis | Statistics were calculated by Bioconductor (v. 3.16) R packages. <br> For NGS data analysis, open source code have been used and listed in the material and methods. Additionally, the entire code for pipeline and data analysis including version control has been deposited in Github https://github.com/Novartis/dms-pipeline/tree/main and Zenodo under record 10418664. |

For manuscripts utilizing custom algorithms or software that are central to the research but not yet described in published literature, software must be made available to editors and reviewers. We strongly encourage code deposition in a community repository (e.g. GitHub). See the Nature Portfolio guidelines for submitting code & software for further information.

## Data

Policy information about availability of data

All manuscripts must include a data availability statement. This statement should provide the following information, where applicable:
- Accession codes, unique identifiers, or web links for publicly available datasets
- A description of any restrictions on data availability
- For clinical datasets or third party data, please ensure that the statement adheres to our policy

All the data have been to SRA with BioProject ID: PRJNA1010676 and can be publicly accessed here ID 1010676 - BioProject - NCBI (nih.gov). All the processed data can additionally be found as supplementary tables and deposited in Zenodo under record 10418664.

## Research involving human participants, their data, or biological material

Policy information about studies with human participants or human data. See also policy information about sex, gender (identity/presentation), and sexual orientation and race, ethnicity and racism.

| | |
|---|---|
| Reporting on sex and gender | not applicable |
| Reporting on race, ethnicity, or other socially relevant groupings | not applicable |
| Population characteristics | not applicable |
| Recruitment | not applicable |
| Ethics oversight | not applicable |

Note that full information on the approval of the study protocol must also be provided in the manuscript.

# Field-specific reporting

Please select the one below that is the best fit for your research. If you are not sure, read the appropriate sections before making your selection.

☒ Life sciences          ☐ Behavioural & social sciences          ☐ Ecological, evolutionary & environmental sciences

For a reference copy of the document with all sections, see nature.com/documents/nr-reporting-summary-flat.pdf

# Life sciences study design

All studies must disclose on these points even when the disclosure is negative.

| | |
|---|---|
| Sample size | Experiments were performed using sample sizes based on standard protocols in the field (n >= 3 for standard cell biology experiments to enable two sided t-tests for normally distributed data). No statistical test was performed to predetermine sample size. |
| Data exclusions | No data were excluded. |
| Replication | All experiments were repeated at least for three times. Detailed information on replicates was available in the figure legends. All attempts to replicate the experiments performed here were successful. |
| Randomization | Samples were processed in a randomized fashion. |
| Blinding | Data acquisition in the studies was conducted in a blinded manner. Data processing was blinded by assigning a random ID constituted of 3 letters and 3 numbers. For analyzing contrasts (e.g. NGS data from different sorted populations or Treated vs. Untreated) the data analyst was informed about the nature of the sample to enable the analysis. |

# Reporting for specific materials, systems and methods

We require information from authors about some types of materials, experimental systems and methods used in many studies. Here, indicate whether each material, system or method listed is relevant to your study. If you are not sure if a list item applies to your research, read the appropriate section before selecting a response.

## Materials & experimental systems

| n/a | Involved in the study |
|-----|----------------------|
| ☐ | ☒ Antibodies |
| ☐ | ☒ Eukaryotic cell lines |
| ☒ | ☐ Palaeontology and archaeology |
| ☒ | ☐ Animals and other organisms |
| ☒ | ☐ Clinical data |
| ☒ | ☐ Dual use research of concern |
| ☒ | ☐ Plants |

## Methods

| n/a | Involved in the study |
|-----|----------------------|
| ☒ | ☐ ChIP-seq |
| ☐ | ☒ Flow cytometry |
| ☒ | ☐ MRI-based neuroimaging |

# Antibodies

| | |
|---|---|
| Antibodies used | Antibodies used:<br>Actin (Millipore, MAB1501; 1:1000 dilution), HA (Cell Signaling, 3724; 1:1000 dilution), ARID1B (Sigma, WH0057492M1, 1:500 dilution), SMARCB1 (Cell signaling, 91735, 1:1000 dilution ), BRG1 (Abcam, ab110641, 1:1000 dilution) and HRP-anti-rabbit and HRP-anti-mouse (Cell Signaling, 1:2500 dilution). |
| Validation | Antibodies were validated by RNAi/CRISPR experiments (western blot upon siRNA or shRNA knockdown or CRISPR KO, data not shown) or cell lines diplaying differential expression.<br>Actin: https://www.merckmillipore.com/CH/de/product/Anti-Actin-Antibody-clone-C4,MM_NF-MAB1501<br>HA: https://www.cellsignal.com/products/primary-antibodies/ha-tag-c29f4-rabbit-mab/3724<br>ARID1B: https://www.sigmaaldrich.com/CH/de/product/sigma/wh0057492m1 and RNAi in Fig S2A<br>SMARCB1: https://www.cellsignal.com/products/primary-antibodies/smarcb1-baf47-d8m1x-rabbit-mab/91735<br>BRG1 (SMARCA4):<br>https://www.abcam.com/products/primary-antibodies/brg1-antibody-epncir111a-ab110641.html |

# Eukaryotic cell lines

Policy information about cell lines and Sex and Gender in Research

| | |
|---|---|
| Cell line source(s) | HEK-293a cells were obtained from Thermo Fisher (R70507) and Cal-51 were obtained from DMSZ (ACC 302).<br>HEK293 ARID1A/B dKO cells were generated by transfecting all-in-one CRISPR plasmids expressing the following sgRNA sequences:  sgARID1B_2 (5'-ACCGTGAGGTGCCAACGTTTAGGT-3') sgARID1B_3 (5'-ACCGAAACTTGATAAGCTTCCTAG-3'), sgARID1B_8 (5'-ACCGGGCACCCCACTATACGCTGG-3'), sgARID1A_2 (5'-ACCGTTGAGATGTCCAAACACCCA-3'), sgARID1A_3 (5'-ACCGGATGTTGGCGAGTGTAACCA-3'), sgARID1A_4 (5'-ACCGCTTGCAACCAACCTCAATGT -3'). |
| Authentication | Cell line identity was confirmed by regular SNP array genotyping |
| Mycoplasma contamination | Cell lines were regularly tested for mycoplasma contamination and cell lines were confirmed to be negative before using for experiments |
| Commonly misidentified lines<br>(See ICLAC register) | No commonly misidentified cell lines were used |

# Flow Cytometry

## Plots

Confirm that:

☒ The axis labels state the marker and fluorochrome used (e.g. CD4-FITC).

☒ The axis scales are clearly visible. Include numbers along axes only for bottom left plot of group (a 'group' is an analysis of identical markers).

☒ All plots are contour plots with outliers or pseudocolor plots.

☒ A numerical value for number of cells or percentage (with statistics) is provided.

## Methodology

| | |
|---|---|
| Sample preparation | Cells were trypsinized and resuspended |
| Instrument | BD Cytoflex LS for analysis and BD FACSAria™ Fusion for FACS sorting |
| Software | CytExpert 2.4.0.28 for analysis and BD FACSDiva™ Software v9.0 for FACS sorting |
| Cell population abundance | Within the "single cells" population, we report % of cells falling into the gate of cells displaying lower GFP levels than mCherry (deviating from the diagonal). |

Gating strategy | Live cells were first gated based on FSC and SSC to exclude debris. After single cells were gated based on FSC-height and FSC-width. Out of the single cells population the cells deviating from the diagonal in the scatter plot comparing  mCherry (in the FL11 channel) and GFP (in the FL1 channel) were quantified. See exemplary figure in Extended Data S2B.

☒ Tick this box to confirm that a figure exemplifying the gating strategy is provided in the Supplementary Information.

