## [Peer Review File · Nature Structural & Molecular Biology]

Peer Review Information

Manuscript Title: Protein destabilization underlies pathogenic missense mutations in ARID1B

Corresponding author name(s): Giorgio Galli, Fabian Sesterhenn

Reviewer Comments & Decisions:

Decision Letter, initial version:
--

Message: 25th Aug 2023

Dear Dr Galli,

Thank you again for submitting your manuscript "Protein destabilization underlies pathogenic missense mutations in ARID1B". I apologise for the delay in responding, which resulted from the difficulty in obtaining suitable referee reports. Nevertheless, we now have comments (below) from the 3 reviewers who evaluated your paper. In light of these reports, we remain interested in your study and would like to see your response to the comments of the referees, in the form of a revised manuscript.

You will see that though all referees appreciate the potential of the approaches used to assess the effects of missense variants in the context of specific syndromes or diseases, in this case CSS, they raise important concerns that necessitate a revised manuscript. More specifically, reviewer #2 raises concerns with respect to the technical validity of the data obtained in the proliferation assays which must be addressed. Reviewer #3 requests that the data pertaining to protein stability are expanded with relevant biochemical and/or biophysical approaches, relevant data are validated in the same or additional cell-lines, as well as investigating the effects of ARID1B mutations in BAF complex stability. Finally, multiple reviewers indicate that certain claims (stability being the only driver for pathogenicity) need to be significantly toned down in the absence of further evidence.

Please be sure to address/respond to all concerns of the referees in full in a point-by-point response and highlight all changes in the revised manuscript text file.

We appreciate the requested revisions are extensive. We thus expect to see your revised manuscript within 6 months. If you cannot send it within this time, please let us know. We

will be happy to consider your revision as long as nothing similar has been accepted for publication at NSMB or published elsewhere. Should your manuscript be substantially delayed without notifying us in advance and your article is eventually published, the received date would be that of the revised, not the original, version.

Reporting Summary:

When submitting the revised version of your manuscript, please pay close attention to our [href="https://www.nature.com/nature-portfolio/editorial-policies/image-integrity">Digital Image Integrity Guidelines.](https://www.nature.com/nature-portfolio/editorial-policies/image-integrity) and to the following points below:

We require deposition of coordinates (and, in the case of crystal structures, structure factors) into the Protein Data Bank with the designation of immediate release upon publication (HPUB). Electron microscopy-derived density maps and coordinate data must be deposited in EMDb and released upon publication. Deposition and immediate release of

NMR chemical shift assignments are highly encouraged. Deposition of deep sequencing and microarray data is mandatory, and the datasets must be released prior to or upon publication. To avoid delays in publication, dataset accession numbers must be supplied with the final accepted manuscript and appropriate release dates must be indicated at the galley proof stage. Please find the complete NRG policies on data availability at <http://www.nature.com/authors/policies/availability.html>.

[Redacted]

Sincerely,

Dimitris Typas
Associate Editor
Nature Structural & Molecular Biology
ORCID: 0000-0002-8737-1319

Referee expertise:

Referee #1: ARID1A/B, integrator, histone PTMs, CSS

Referee #2: Deep mutational scanning

Referee #3: ARID1A/B complexes, synthetic lethality screens in cancer

Reviewers' Comments:

Reviewer #1:

Remarks to the Author:

Components of the BAF complex are mutated in a significant portion of solid tumors and in a set of developmental syndromes, including Coffin-Siris (CSS). The ARID1A/B subunits of canonical BAF, frequently targeted by pathogenic mutations, are the largest components of the complex, and are likely implicated in targeting the nucleosome remodeling activity of BAF to specific promoters and enhancers. Defects in ARID1B compromise the pattern of chromatin accessibility in the developing neural crest leading to a broad array of morphological and neurological defects.

This work employs deep mutational scanning of ARID1B to ultimately argue that most, if not all, pathogenic missense mutations of ARID1B are essentially loss-of-function mutations, in that they result in protein instability/misfolding. The authors elegantly combine a protein stability assay using a fluorescent GFP reporter (and an internal control) with a cellular proliferation assay that exploits the synthetic lethality of ARID1B with ARID1A.

The manuscript is well-written, technically sound, and all experiments are well-controlled. Findings are interesting, albeit descriptive, and the principles could be translated to ARID1A and its mutational landscape in cancer.

Reviewer #2:

Remarks to the Author:

This manuscript presents a pair of deep mutational scanning data sets for the ARID1B protein, one for the effect of variants in a proliferation assay and the other for the effect of variants on stability. The main claim of the manuscript is that pathogenic variants act by disrupting stability. Overall, the manuscript is clearly written and the figures are easy to interpret. Although brief, the manuscript does a good job of presenting the data and discussing conclusions.

My main worry about this paper is that the proliferation assay does not appear to be working. The distribution of variant effects is basically symmetric, with about as many variants causing gain of proliferation as loss of proliferation. This is a really strange result, as, for most proteins (and for the stability assay presented in the paper) the overwhelming majority of variants are either like wild type or else cause loss of function. The authors perform some individual validation of pathogenic variants in the proliferation assay and here, too, the effects look minimal. The authors should either present data clarifying that the proliferation assay actually worked or else they should explain that it did not work that well. They would also then have to moderate their claims about stability loss being the "only" driver of pathogenicity. Provided this central issue can be resolved, I think the manuscript is a strong contribution.

Major comments:

The manuscript appears to be missing a data availability statement. The functional data should be deposited into MaveDB, Zenodo or some other versioned repository. The code for analysis should be available on GitHub or similar.

The authors identify one missense variant, E2029K, that is pathogenic but not destabilizing in their GFP-based stability assay. They nicely confirm, using western blots, that this result is not an artifact of the stability assay. What do they think is going on with this variant?

Somewhere, the authors should add a supplemental figure showing the replicate correlations for each assay. This might help address the points I raised earlier about the proliferation assay.

Did the authors' library contain stop codons and synonymous variants? If so, they should add a supplemental figure showing how these variants behave relative to missense variants.

Several other DMS'es for protein stability have been performed. The authors should comment on how their results relate.

Minor comments:

Line 62 - The authors claim that most mutations do not impact function or stability. They should give numbers or fractions here, since it is hard to tell from the (overplotted) SF4.

Line 64-66 - The authors claim that no mutations caused loss of proliferation without loss of stability. But, Fig 1C seems to contradict this, in that there are some points in the lower left corner of each plot. The authors should clarify.

Line 65 - The distribution of mutant effects for the proliferation screen looks pretty strange. Basically, it seems like about half the variants depress growth and the other half increase growth. This is not a typical result from a DMS and, to me, looks like the proliferation assay is mostly noise.

Line 75 - It's not clear to me what "differential representation analysis" means. Is that just log2 fold change?

Line 91/Fig 2c - It's very hard to find E2029K in Figure 2C because it is near the dotted line. Perhaps if all the points were bigger or their color more striking (e.g. black) it would be easier.

Line 92 - "assayed" is probably supposed to be "assayed"

Line 94 - The authors test pathogenic variants in their growth assay individually. The results look pretty underwhelming to me, in that the difference between the DOX and NODOX for the benign variants doesn't look to be too different than for the pathogenic variants.

Figure 1a - The authors should state when they accessed ClinVar to generate the plot.

SF4 - Presumably the colored points indicate statistically significant differences. This should be stated on the plot/in the legend.

Fig 2a - The log2fc_sensor legend should be changed to match the rest of the figures. Also, and this is an editorial comment, most DMS papers use a blue = loss of function to red = gain of function (with white being normal function) scale. I suggest the authors conform to this standard.

Reviewer #3:

Remarks to the Author:

The manuscript by Mermet-Meillon et al. entitled "Protein destabilization underlies pathogenic missense mutations in ARID1B" provides insights into the functional impact of several missense mutations located within the ARID1B EHD2 domain in orthogonal readouts of protein stability and cancer cell proliferation. By creating a pool of 8960 ARID1B point mutations, they found that the majority of these mutations had no effect on either protein stability or cell proliferation. However, for the mutations that did elicit decreased protein stability, there was a consistent reduction of cell proliferation. Additionally, they discovered that no single mutation at the interface between ARID1B and BRG1 affected ARID1B stability or led to any phenotypic changes, which indicates that the primary mechanism underlying pathogenic missense mutations in ARID1B is associated with protein destabilization rather than disruptions in protein-protein interactions. Furthermore, they connected pathogenic mutations found in Coffin-Siris Syndrome (CSS) to ARID1B destabilization, reinforcing the clinical relevance and its potential implications for understanding the molecular basis of CSS. Overall, the manuscript is of potential interest; however, the following comments should be addressed.

Major concerns:

1. The authors proposed that single mutations inhibiting ARID1B function are due to protein destabilization rather than other mechanisms such as blocking protein-protein interactions (PPI). They also found that no single mutation at the interface between ARID1B and BRG1 affected ARID1B stability or inhibited cancer cell proliferation. It would be compelling to assess if mutations at the interface between ARID1B and BRG1 impair their interaction by immunoprecipitation assays. Furthermore, it also would be informative if the authors could explore whether point mutations of ARID1B affect BAF complex stability.
2. It would be beneficial if the authors could include biochemical or biophysical assays such as Differential Scanning Calorimetry (DSC) or Circular Dichroism (CD) spectroscopy, to compare the protein stability of ARID1B with or without single mutations identified in the functional assays. Likewise, the regions of ARID1B that the authors suggest are highly destabilized should be shown at protein levels by running western blot assays. Also, for these mutations the authors should do colony formation assays depicting any change in proliferation.
3. While the authors connected pathogenic mutations observed in CSS patients to ARID1B destabilization, caution should be taken when claiming in the Abstract that "...protein destabilization is the major mechanisms elicited by pathogenic missense mutations in Coffin Siris Syndrome patients." Notably, the molecular mechanism they identified in ARID1A-deficient cancer cell line Cal51 may not necessarily apply for CSS patients.
4. Likewise, the authors have made use of "pathogenic missense mutation" terminology which is inappropriate. Since, the CSS is a syndrome and not a disease. Hence the authors should try using only missense mutation in their manuscript.
5. The assays for proliferation and measuring mCherry/GFP ratio appear to be conducted in different cell lines (HEK293A and Cal51 cells). At least for those with pathological relevance, this should be validated in Cal51 cells. To increase the rigor of the findings, the authors should consider repeating the experiments in another ARID1A-mutated cell line.

Minor Concerns:

1. The authors included HA-ARID1B (del BC-box) as a control to demonstrate decreased protein stability and proliferation. However, the Western blot results indicate an abundance of the protein level (Ext Fig 2a). Moreover, in the group of HA-ARID1B (del BC-

box), the endogenous ARID1B level was cleared off, whereas this was not observed in HA-ARID1B (del EHD2). To enhance clarity, it would be beneficial if the authors could label the BC-box in the diagram depicting ARID1B domains.

2. The strategy of the proliferation assay that the authors have implemented is quite unclear. If the authors have just constructed their libraries encompassing the EHD2 domain of ARID1B, then they should not have observed any proliferation in ARID1Amut expressing Dox-inducible shARID1B.

3. The manuscript is dense and at times hard to follow.

Author Rebuttal to Initial comments

Referee expertise:

Referee #1: ARID1A/B, integrator, histone PTMs, CSS

Referee #2: Deep mutational scanning

Referee #3: ARID1A/B complexes, synthetic lethality screens in cancer

Reviewers' Comments:

Reviewer #1:

Remarks to the Author:

Components of the BAF complex are mutated in a significant portion of solid tumors and in a set of developmental syndromes, including Coffin-Siris (CSS). The ARID1A/B subunits of canonical BAF, frequently targeted by pathogenic mutations, are the largest components of the complex, and are likely implicated in targeting the nucleosome remodeling activity of BAF to specific promoters and enhancers. Defects in ARID1B compromise the pattern of chromatin accessibility in the developing neural crest leading to a broad array of morphological and neurological defects.

This work employs deep mutational scanning of ARID1B to ultimately argue that most, if not all, pathogenic missense mutations of ARID1B are essentially loss-of-function mutations, in that they result in protein instability/misfolding. The authors elegantly combine a protein stability assay using a fluorescent GFP reporter (and an internal control) with a cellular proliferation assay that exploits the synthetic lethality of ARID1B with ARID1A.

The manuscript is well-written, technically sound, and all experiments are well-controlled.

Findings are interesting, albeit descriptive, and the principles could be translated to ARID1A and its mutational landscape in cancer.

We thank the reviewer for appreciating our work.

Reviewer #2:

Remarks to the Author:

This manuscript presents a pair of deep mutational scanning data sets for the ARID1B protein, one for the effect of variants in a proliferation assay and the other for the effect of variants on stability. The main claim of the manuscript is that pathogenic variants act by disrupting stability. Overall, the manuscript is clearly written and the figures are easy to interpret. Although brief, the manuscript does a good job of presenting the data and discussing conclusions.

My main worry about this paper is that the proliferation assay does not appear to be working. The distribution of variant effects is basically symmetric, with about as many variants causing gain of proliferation as loss of proliferation. This is a really strange result, as, for most proteins (and for the stability assay presented in the paper) the overwhelming majority of variants are either like wild type or else cause loss of function. The authors perform some individual validation of pathogenic variants in the proliferation assay and here, too, the effects look minimal. The authors should either present data clarifying that the proliferation assay actually worked or else they should explain that it did not work that well. They would also then have to moderate their claims about stability loss being the “only” driver of pathogenicity. Provided this central issue can be resolved, I think the manuscript is a strong contribution.

We thank the reviewer for appreciating the clarity and quality of the manuscript as well as providing insightful comments.

This prompted us to investigate further the validity of our proliferation assay and how those results relate to the results of the stability sensor.

First, we observed significant clustering among individual samples suggesting that we are likely detecting true signal, with top hits sharing consistent behavior across different replicates. See Figure 9 and 10 for reviewer within the specific major point below.

We then sought to additionally evaluate the reliability of the proliferation assay by comparing mutants that affect stability and proliferation (called “prolif” for simplicity) with mutants that, surprisingly, affect only protein stability but not cancer cells proliferation (called “inert”). See boxes in scatter plots below.

Interestingly, we observed a small but significant higher level of destabilization for the “prolif” mutants vs. the “inert” ones mostly driven by a higher proportion (more homogeneous distribution) of highly destabilizing mutations in the “prolif” subset. See below.

Figure 1 for reviewer

We believe this is due to an enrichment of more deleterious mutations (e.g. introduction of charges) in hydrophobic residues, which might result in a more severe perturbation of the protein core. See below.

Figure 2 for reviewer

Nevertheless, these differences are likely not sufficient to explain the major differences in phenotypic outcome between “inert” and “prolif” mutants. Thereby we performed extensive validation on 13 selected mutants (6-7 from each category) for residues that reside at 10-15Å distance to BRG1 while attempting to balance type of mutations.

Figure 3 for reviewer

We first indeed validated by sensor assay in two different cell lines (and western blot, see point below about co-IP) that the selected mutants all destabilize the protein with the “prolif” set tending to have higher destabilization potential according to the cell line or technique used. See below, these data have now been introduced also into the manuscript as Fig. 1D and Extended data Fig. 5b.

Figure 4 for reviewer

We also validated the proliferation screen by using colony formation assay, reproducing the screening data (i.e. “prolif” mutants consistently fail to rescue Cal51 cells proliferation upon depletion of endogenous ARID1B). These data have been included as Fig. 1e.

Figure 5 for reviewer

We additionally validated the proliferation data using an incuCyte-based cell growth assay, which, again, revealed that only “prolif” are unable to sustain cell proliferation (note delta growth in DOX vs. NODOX condition for “orange mutants”). These data have been included in Extended data Fig. 5c.

Figure 6 for reviewer

Having established the reproducibility of the screening assays, we then sought to understand if complex composition could be underlying the differences in proliferative phenotypes between “prolif” and “inert” mutants. Co-immunoprecipitation assay between HA-tagged ARID1B constructs and key subunits of SWI/SNF revealed the complete inability of “prolif” mutants to being incorporated in the complex (see below, these data have been added as Fig. 1f). We thereby here postulate that, while destabilization might be sufficient to perturb the developmental phenotype in CSS patients (indeed most CSS pathogenic mutants are destabilized but do not elicit antiproliferative effects), to elicit a phenotype in ARID1A mutant cells by perturbing ARID1B functions, complex disassembly needs to be achieved on top of destabilization. This raises the bar for therapeutic targeting of ARID1B in ARID1A mutant cancers.

Figure 7 for reviewer

Major comments:

The manuscript appears to be missing a data availability statement. The functional data should be deposited into MaveDB, Zenodo or some other versioned repository. The code for analysis should be available on GitHub or similar.

We added a data availability statement. All the data have been to SRA with BioProject ID: PRJNA1010676 and can be publicly accessed here [ID_1010676 - BioProject - NCBI \(nih.gov\)](https://www.ncbi.nlm.nih.gov/bioproject/1010676). All the processed data are in supplementary material for easy public access. We also added a code availability statement. It is worth noticing that we are not using any proprietary software or any particular package, thereby we provide step-by-step information about our computational analyses directly in the Methods section. This should allow any computational biologist to reproduce our findings

The authors identify one missense variant, E2029K, that is pathogenic but not destabilizing in their GFP-based stability assay. They nicely confirm, using western blots, that this result is not an artifact of the stability assay. What do they think is going on with this variant?

Unfortunately, we are unable to rationalize the annotated pathogenic effect of E2029K. E2029 is a surface exposed residue at the N terminus of one of the central helices in ARID1b, facing SMARCB1. An initial hypothesis was that E2029K affects the interaction with SMARCB1, we could not confirm such effect by IP. Likewise, as pointed out, multiple assays have confirmed that E2029K does not affect ARID1B levels. Considering the paucity of evidence supporting the pathogenicity of this mutation according to ClinVar annotation (see snapshot below), we believe that this could be either a misannotation in ClinVar or a cell type specific effect on BAF complex formation that escapes detection in our assay system.

Submitted interpretations and evidence ?

Interpretation (Last evaluated)	Review status (Assertion criteria)	Condition (Inheritance)	Submitter	More information
Likely pathogenic (Jun 04, 2018)	criteria provided, single submitter (GeneDx Variant Classification (06012015)) Method: clinical testing	- Not Provided Affected status: yes Allele origin: germline	GeneDx Accession: SCV000569966.3 First in ClinVar: Apr 27, 2017 Last updated: Apr 27, 2017	[ ]
Comment: The E2042K variant in the ARID1B gene has not been reported previously as a pathogenic variant, nor as a benign variant, to our knowledge. The E2042K variant is not observed in large population cohorts (Lek et al., 2016). The E2042K variant is a non-conservative amino acid substitution, which is likely to impact secondary protein structure as these residues differ in polarity, charge, size and/or other properties. In-silico analyses, including protein predictors and evolutionary conservation, support a deleterious effect. We interpret E2042K as a likely pathogenic variant. (less)				

Figure 8 for reviewer

Somewhere, the authors should add a supplemental figure showing the replicate correlations for each assay. This might help address the points I raised earlier about the proliferation assay.

We thank the reviewer for this suggestion. We performed PCA analysis to analyze the variance across replicates of proliferation screen and observed correct clustering for both Pool4 and Pool5 (see scatter plot of the PCA below).

Fig. 9 for

reviewer

Similarly hierarchical clustering of such samples similarly provided correct grouping, further supporting the robustness of signal of the proliferation screen across replicates (see heatmap with dendrograms below).

Figure 10 for reviewer

Did the authors' library contain stop codons and synonymous variants? If so, they should add a supplemental figure showing how these variants behave relative to missense variants.

Unfortunately, we did not add stop codons or synonymous variants to the libraries. As a surrogate, we are reporting here the scoring of “proline mutants in helical regions” (predicted to have high chance of scoring) vs. “mutations of Isoleucines into Leucines and viceversa” (predicted to be mostly neutral mutations). We do observe indeed striking differences in scoring between these two categories, in line with wide dynamic range of our sensor assay.

Figure 11 for reviewer

Several other DMS'es for protein stability have been performed. The authors should comment on how their results relate.

We thank the reviewer for this comment. In the text we added reference to papers using different DMS setups that mention that perturbation to the hydrophobic core leads to destabilization or loss of function, including two VAMP-seq papers (ref 10: Matreyek et al., Nat Genetics, 2018 and ref 14: Amorosi CJ et al., AJHG, 2021).

Minor comments:

Line 62 - The authors claim that most mutations do not impact function or stability. They should give numbers or fractions here, since it is hard to tell from the (overplotted) SF4.

Thanks for this suggestion and apologies for overplotting. We marked the number of scoring hits (up- or down-regulated) relative to total number of variants in the library.

Line 64-66 - The authors claim that no mutations caused loss of proliferation without loss of stability. But, Fig 1C seems to contradict this, in that there are some points in the lower left corner of each plot. The authors should clarify.

We apologize for this inaccuracy. Indeed, there are few mutants affecting proliferation without stability loss (i.e. lower left corner dots), albeit these mutants score with lower statistical significance. We modified the text accordingly.

Line 65 - The distribution of mutant effects for the proliferation screen looks pretty strange. Basically, it seems like about half the variants depress growth and the other half increase growth. This is not a typical result from a DMS and, to me, looks like the proliferation assay is mostly noise.

We thank the reviewer for this insight. It is true that by employing DESeq2 for differential analysis, which is based on TMM (median) normalized values, we potentially enhanced the “centering” of the volcano plot. However, the “upregulated” mutants display lower significance values. Most importantly, we believe in the validity of the screen based on: the PCA analysis above, the reproducibility of data of the proliferation assay of overall 42 mutants and the distinct pattern of complex incorporation observed for scoring mutants.

Line 75 - It's not clear to me what “differential representation analysis” means. Is that just log₂ fold change?

To identify hits, we simply use DeSeq2, which is widely applied to identify differential gene expression (i.e. for RNA-seq), differential occupancy (i.e. for ChIP-seq) and, in this case, for differential abundance of mutants across different samples (i.e. their representation in the library). DeSeq2 statistical framework returns a log₂ fold change and pvalue adjusted for multiple testing. To avoid confusions we refer to the analysis as “differential analysis” or “differential abundance analysis”. See improved methods section for further details.

Line 91/Fig 2c - It's very hard to find E2029K in Figure 2C because it is near the dotted line. Perhaps if all the points were bigger or their color more striking (e.g. black) it would be easier.

We apologize for this. To highlight that E2029K is indeed in the category “likely pathogenic”, we darkened the color of the dots and added a very visible label.

Line 92 - “assayed” is probably supposed to be “assayed”

Thank you for spotting this typo. We corrected it.

Line 94 - The authors test pathogenic variants in their growth assay individually. The results look pretty underwhelming to me, in that the difference between the DOX and NODOX for the benign variants doesn't look to be too different than for the pathogenic variants.

We apologize for not explaining appropriately our data. This comment provides us an opportunity to present our data in a more exhaustive way. As mentioned above, we demonstrated that we could reproduce data from our screens by selecting destabilizing +/- anti-proliferative mutants.

When selecting instead mutants annotated in ClinVar according to "pathogenicity", we first demonstrate that all (except E2029K) pathogenic mutations destabilize the protein compared to benign mutations, while "uncertain" mutations exert a variable outcome. We now validate thereby our screen using our reporter assay also in Cal51 cells on top of HEK293A and by WB.

Validation of stability screen by FACS-based reporter assay

Figure 12 for reviewer

Thereby, we established that destabilization is likely associated with the "pathogenicity" of mutations, as reported by ClinVar. However, as mentioned above during the validation of the "prolif" mutants, we do not expect that also an effect on cancer cell proliferation should be necessarily associated with mutations found in neurodevelopmental syndrome patients. Indeed, as correctly pointed out by the reviewer, we do not observe an effect on Cal51 proliferation

(proliferation screen, CFA or Incucyte assay) or complex assembly (by Co-IP) by ClinVar mutants, except for L2073R.

Validation of proliferation screen by Colony Formation assay

Fig. 13

for review

Figure 14 for reviewer

Figure 15 for reviewer

Figure 1a - The authors should state when they accessed ClinVar to generate the plot.

We added the release date (October 2022) in figure legend.

SF4 - Presumably the colored points indicate statistically significant differences. This should be stated on the plot/in the legend.

We added this information in the figure legend.

Fig 2a - The log2fc_sensor legend should be changed to match the rest of the figures. Also, and this is an editorial comment, most DMS papers use a blue = loss of function to red = gain of function (with white being normal function) scale. I suggest the authors conform to this standard.

We thank the reviewer for your editorial comment. We fixed the figure legend and changed all heatmaps and structure figures using the blue = loss, red = gain palette.

Reviewer #3:

Remarks to the Author:

The manuscript by Mermet-Meillon et al. entitled “Protein destabilization underlies pathogenic missense mutations in ARID1B” provides insights into the functional impact of several missense mutations located within the ARID1B EHD2 domain in orthogonal readouts of protein stability and cancer cell proliferation. By creating a pool of 8960 ARID1B point mutations, they found that the majority of these mutations had no effect on either protein stability or cell proliferation. However, for the mutations that did elicit decreased protein stability, there was a consistent reduction of cell proliferation. Additionally, they discovered that no single mutation at the interface between ARID1B and BRG1 affected ARID1B stability or led to any phenotypic changes, which indicates that the primary mechanism underlying pathogenic missense mutations in ARID1B is associated with protein destabilization rather than disruptions in protein-protein interactions. Furthermore, they connected pathogenic mutations found in Coffin-Siris Syndrome (CSS) to ARID1B destabilization, reinforcing the clinical relevance and its potential implications for understanding the molecular basis of CSS. Overall, the manuscript is of potential interest; however, the following comments should be addressed.

We thank the reviewer for appreciating our manuscript and providing an insightful and thoughtful revision.

Major concerns:

1. The authors proposed that single mutations inhibiting ARID1B function are due to protein destabilization rather than other mechanisms such as blocking protein-protein interactions (PPI). They also found that no single mutation at the interface between ARID1B and BRG1 affected ARID1B stability or inhibited cancer cell proliferation. It would be compelling to assess if mutations at the interface between ARID1B and BRG1 impair their interaction by immunoprecipitation assays. Furthermore, it also would be informative if the authors could explore whether point mutations of ARID1B affect BAF complex stability.

We thank the reviewer for this insightful comment. As reported in Extended data Fig. 5a in the manuscript (and see major remark by Reviewer 2), when considering a set of mutants at 10-15Å distance from BRG1, we have been able to clarify potential features underlying differences in mutations affecting protein stability vs. destabilizing mutations additionally affecting cancer cell proliferation.

Indeed we confirmed that “only destabilizing” mutations (called “inert” for simplicity) decrease ARID1B protein levels, but do not affect cancer cell proliferation (Fig 1d, 1e, Extended Data Fig. 5b, 5c in revised manuscript or look at Figure 3-7 for reviewers). Nevertheless, protein destabilization is associated with mutations annotated in ClinVar as “pathogenic” indicating that differences in protein stability can perturb neurodevelopmental processes.

Instead, to perturb the proliferation of ARID1A^{mut} cancer cells, “prolif” mutations, not only are destabilizing, but they also fail to be incorporated into SWI/SNF complex (Fig. 1f in the revised manuscript or look at Figure 3-7 for reviewers). This suggests that perturbing ARID1B to exploit the synthetic lethal paradigm in cancer requires a higher bar such as complex disruption.

2. It would be beneficial if the authors could include biochemical or biophysical assays such as Differential Scanning Calorimetry (DSC) or Circular Dichroism (CD) spectroscopy, to compare the protein stability of ARID1B with or without single mutations identified in the functional assays. Likewise, the regions of ARID1B that the authors suggest are highly destabilized should be shown at protein levels by running western blot assays. Also, for these mutations the authors should do colony formation assays depicting any change in proliferation.

We attempted to express and purify multiple ARID1B constructs in E. Coli (never obtained soluble expression), mammalian cells (never obtained sufficient yields and purity for downstream assays) and Insect cells (successfully obtaining wild type ARID1B EHD2 domain).

However, when we started generating a variety of mutants (mutations, truncations etc.), we observed that the protein is very sensitive to modifications. Indeed, we failed to obtain sufficient soluble expression for purification purposes for several mutants/truncations, suggesting that indeed perturbations in the hydrophobic core might prevent expression also with Baculovirus system.

Thereby, following reviewer's suggestion, we extensively complemented our validation studies for two sets of mutants ("inert vs. proliferating" and "ClinVar annotated") with several measurements: Stability sensor in HEK293A cells and Cal51, Western blot with Co-IPs, colony formation assay, Incucyte-based growth assay (Figure 3-7 for reviewers). These data have been added to the manuscript at Fig. 1d-e-f and Extended data Fig. 5.

3. While the authors connected pathogenic mutations observed in CSS patients to ARID1B destabilization, caution should be taken when claiming in the Abstract that "...protein destabilization is the major mechanisms elicited by pathogenic missense mutations in Coffin Siris Syndrome patients." Notably, the molecular mechanism they identified in ARID1A-deficient cancer cell line Cal51 may not necessarily apply for CSS patients.

We apologize for the inaccuracy. We believe our new data help to clarify that indeed there are different molecular mechanisms underlying cancer cell proliferation vs. association with CSS pathogenicity. We rephrased the abstract to be more precise.

4. Likewise, the authors have made use of "pathogenic missense mutation" terminology which is inappropriate. Since, the CSS is a syndrome and not a disease. Hence the authors should try using only missense mutation in their manuscript.

We do agree that the term "pathogenic" might be unprecise. However, we decided to use such term to align to the ones used in the ClinVar annotation for ease of the broad readership. We redacted the term in some occurrences where possible.

5. The assays for proliferation and measuring mCherry/GFP ratio appear to be conducted in different cell lines (HEK293A and Cal51 cells). At least for those with pathological relevance, this should be validated in Cal51 cells. To increase the rigor of the findings, the authors should consider repeating the experiments in another ARID1A-mutated cell line.

We thank the reviewer for this comment. As mentioned above we extended our validation package for several mutations including the analysis of protein stability in Cal51 cells to allow consistency with the cell proliferation assay. Importantly, we see very similar results in the sensor assay between HEK293A and Cal51. These data have been added to Fig. 1d, 2e, Extended data Fig. 5b, 9a in the revised manuscript.

Minor Concerns:

1. The authors included HA-ARID1B (del BC-box) as a control to demonstrate decreased protein stability and proliferation. However, the Western blot results indicate an abundance of the protein level (Ext Fig 2a). Moreover, in the group of HA-ARID1B (del BC-box), the endogenous ARID1B level was cleared off, whereas this was not observed in HA-ARID1B (del EHD2). To enhance clarity, it would be beneficial if the authors could label the BC-box in the diagram depicting ARID1B domains.

We thank the reviewer for noticing the inconsistencies and apologize for the confusion. The labels of the last two samples were swapped. Indeed the Δ EHD2 domain runs at a lower molecular weight (we removed around 60 KDa) and it displays high expression. On the contrary the Δ BC-Box retains similar molecular weight (overlapping with endogenous in a WB using ARID1B antibody) and displays lower abundance. We corrected this mistake in Extended data Figure 2.

2. The strategy of the proliferation assay that the authors have implemented is quite unclear. If the authors have just constructed their libraries encompassing the EHD2 domain of ARID1B, then they should not have observed any proliferation in ARID1A^{mut} expressing Dox-inducible shARID1B.

We apologize for the unclarity. The proliferation assay and the stability sensor assay use two different libraries. To enable functional rescue, we cloned the mutant cassette into an HA-tagged Full Length ARID1B cDNA to enable. Conversely, for degradation sensor assay, we used only the EHD2 domain to allow space in the lentiviral vector to include the mCherry-P2A-GFP cassette. We updated the diagram in Figure 1B to improve the clarity.

3. The manuscript is dense and at times hard to follow.

We thank the reviewer for the comment. We have revised text and figures to improve clarity, while trying to follow the format of brief communication format, which does not allow for extensive elaboration.

Decision Letter, first revision:

Message: Our ref: NSMB-BC47456B

7th Dec 2023

Dear Dr. Galli,

Thank you for submitting your revised manuscript "Protein destabilization underlies pathogenic missense mutations in ARID1B" (NSMB-BC47456B). It has now been seen by the original referees and their comments are below. All three reviewers, including reviewer #1 who had no further comments for the authors, find that the paper has improved in revision, and therefore we'll be happy to accept it in principle in Nature Structural & Molecular Biology, pending minor revisions to satisfy the referees' final requests (most importantly depositing custom code to a public repository) and to comply with our editorial and formatting guidelines.

We are now performing detailed checks on your paper and will send you a checklist detailing our editorial and formatting requirements in about two weeks. Please do not upload the final materials and make any revisions until you receive this additional information from us.

To facilitate our work at this stage, it is important that we have a copy of the main text as a word file. If you could please send along a word version of this file as soon as possible, we would greatly appreciate it; please make sure to copy the NSMB account (cc'ed above).

Sincerely,

Dimitris Typas
Associate Editor
Nature Structural & Molecular Biology
ORCID: 0000-0002-8737-1319

Reviewer #2 (Remarks to the Author):

The authors have addressed most of my concerns in their revision. The remaining point,

which I feel is very important, is that the reviewers should make their code data available in a versioned repository. While a supplemental table is a reasonable starting point, it is insufficient.

They should create a GitHub repository for their code and also deposit the processed data in Zenodo or some other location. These steps are necessary to ensure that their work is reproducible and the data useable in the future.

Reviewer #3 (Remarks to the Author):

In this revised manuscript the authors have answered all of my previous concerns and I recommend the publication of the manuscript at NSMB. Congrats to the authors. Rugang Zhang

Author Rebuttal, first revision:

Reviewer's comments

Reviewer #2 (Remarks to the Author):

The authors have addressed most of my concerns in their revision. The remaining point, which I feel is very important, is that the reviewers should make their code data available in a versioned repository. While a supplemental table is a reasonable starting point, it is insufficient.

They should create a GitHub repository for their code and also deposit the processed data in Zenodo or some other location. These steps are necessary to ensure that their work is reproducible and the data useable in the future.

We thank the reviewer for the insightful comments and apologize it took long to get our code deposited. We now deposited the code for our pipeline in Github <https://github.com/Novartis/dms-pipeline/tree/main> and the entire downstream code (with version control) and all tables in Zenodo under record 10418664.

Reviewer #3 (Remarks to the Author):

In this revised manuscript the authors have answered all of my previous concerns and I recommend the publication of the manuscript at NSMB. Congrats to the authors. Rugang Zhang

We thank you for the insightful revision of our paper

Final Decision Letter:**Message** 18th Jan 2024

:

Dear Dr. Galli,

We are now happy to accept your revised paper "Protein destabilization underlies pathogenic missense mutations in ARID1B" for publication as a Brief Communication in Nature Structural & Molecular Biology.

Your paper will be published online soon after we receive proof corrections and will appear in print in the next available issue. You can find out your date of online publication by contacting the production team shortly after sending your proof corrections.

You may wish to make your media relations office aware of your accepted publication, in

case they consider it appropriate to organize some internal or external publicity. Once your paper has been scheduled you will receive an email confirming the publication details. This is normally 3-4 working days in advance of publication. If you need additional notice of the date and time of publication, please let the production team know when you receive the proof of your article to ensure there is sufficient time to coordinate. Further information on our embargo policies can be found here:
<https://www.nature.com/authors/policies/embargo.html>

Please note that *Nature Structural & Molecular Biology* is a Transformative Journal (TJ). Authors may publish their research with us through the traditional subscription access route or make their paper immediately open access through payment of an article-processing charge (APC). Authors will not be required to make a final decision about access to their article until it has been accepted. Find out more about Transformative Journals <https://www.springernature.com/gp/open-research/transformative-journals>

Sincerely,

Dimitris Typas
Associate Editor
Nature Structural & Molecular Biology
ORCID: 0000-0002-8737-1319